# Strongly deleterious mutations influence reproductive output and longevity in an endangered population

Malin Hasselgren [1] ✉, Nicolas Dussex [1,2,3], Johanna von Seth[1,2,3], Anders Angerbjörn [1], Love Dalén [1,2,3] & Karin Norén[1]

Inbreeding depression has been documented in various fitness traits in a wide range of species and taxa, however, the mutational basis is not yet well understood. We investigate how putatively deleterious variation influences fitness and is shaped by individual ancestry by re-sequencing complete genomes of 37 individuals in a natural arctic fox (*Vulpes lagopus*) population subjected to both inbreeding depression and genetic rescue. We find that individuals with high proportion of homozygous loss of function genotypes (LoFs), which are predicted to exert a strong effect on fitness, generally have lower lifetime reproductive success and live shorter lives compared with individuals with lower proportion of LoFs. We also find that juvenile survival is negatively associated with the proportion of homozygous missense genotypes and positively associated with genome wide heterozygosity. Our results demonstrate that homozygosity of strongly and moderately deleterious mutations can be an important cause of trait specific inbreeding depression in wild populations, and mark an important step towards making more informed decisions using applied conservation genetics.

Many once large and continuous populations today persist as small and isolated subpopulations that are threatened by extinction[1,2]. Reduced genetic variation due to genetic drift and inbreeding is considered a major threat to long-term persistence of fragmented populations[3] and it has long been discussed how to best maintain genetic variation in wild populations[4–7]. Recently, it has been increasingly debated whether it is more beneficial to maximize genetic variation or to minimize deleterious variation in small populations[8–10]. Although inbreeding depression has been of scientific interest to biologists for over a century[11,12], the underlying mechanisms remain challenging to study and little is known about how deleterious variation across the genomic landscape influences fitness in wild populations. For example, it is not well understood how many alleles contribute to inbreeding depression[13]. The number, strength, and distribution of deleterious variants could in turn influence the efficiency of removal of deleterious variation through natural selection (i.e., purging) from at risk populations[14].

The traditional view on the most suitable approach to reduce inbreeding has been to increase genetic variation through gene flow from unrelated individuals from large and outbred source populations with high genetic diversity[15,16], however this has recently been widely debated[8,10,17–19]. It was recently proposed that gene flow from large populations with high genetic diversity into small, inbred populations can result in elevated extinction risk through increasing the mutational load (i.e., genetic burden of a population resulting from accumulated deleterious mutations) due to introduction of a large number of recessive harmful alleles naturally harbored by large populations[10,20]. On the other hand, outbred immigrants from a large source population are more likely to have higher fitness and adaptive potential[8] as well as lower drift load[9]. Whether it is more efficient to use large source populations or small to medium sized ones to mitigate inbreeding depression depends on the underlying mechanism of inbreeding depression, for instance whether it is driven by many small-effect

[1]Department of Zoology, Stockholm University, Stockholm, Sweden. [2]Centre for Palaeogenetics, Svante Arrhenius väg 20C, Stockholm, Sweden. [3]Department of Bioinformatics and Genetics, Swedish Museum of Natural History, Stockholm, Sweden. ✉e-mail: malin.hasselgren@zoologi.su.se

mutations or fewer ones of large effect. In domesticated populations, pedigree-based approaches have often revealed identity-by-descent of a small subset of strongly deleterious alleles to be involved in inbreeding depression[21–23]. A limited number of pedigree-based approaches have also been used to demonstrate lethal conditions (such as blindness and chondrodystrophy) to be caused by single-locus recessive alleles[24,25]. However, it is only with recent genomic techniques that genetic causes of inbreeding depression can be studied more in depth across the entire genome.

The advance of genomic approaches is rapidly improving the possibilities to study the genetic basis of inbreeding depression in wild populations. However, although genomics is a powerful tool for conservation of threatened wild populations, few studies have investigated the consequences of putatively deleterious mutations on individual fitness. In Soay sheep (*Ovis aries*) and red deer (*Cervus elaphus*), inbreeding depression was mainly due to many alleles of small effect[26,27]. In Soay sheep[26] and in the hihi (*Notiomystis cincta*)[28], inbreeding depression in some traits could also be associated with a small number of SNPs through genome wide association scans. Stoffel et al.[29] recently found an indirect link between mutational load and fitness, and demonstrated through population genetic simulations that long runs of homozygosity (ROH) were enriched with deleterious mutations and that long ROH lowered survival disproportionately more than short ROH. However, a direct link between individual fitness traits and deleterious variation in wild populations has to our knowledge not yet been established.

The endangered arctic fox (*Vulpes lagopus*) population in Scandinavia is a suitable system to study deleterious mutations and their effect on fitness. The population is closely monitored on an individual level and is known to suffer from inbreeding depression[30]. Moreover, genetic rescue was documented following the arrival of three outbred male foxes released from a captive breeding station in Norway, which led to higher fitness in F1 hybrids between immigrants and natives[31]. However, it was recently revealed that the rescue effect only lasted for one single generation, as no elevated fitness was documented for F2 and F3 descendants of immigrants[32] and long runs of homozygosity were found in some F2 genomes[33]. However, it is not known how individual ancestry shapes deleterious variation. Further, a link between deleterious variation and key fitness traits has not been established.

Here, we study the connections between genomic variation, immigration and fitness. Specifically we (i) investigate whether the proportion of predicted deleterious mutations in homozygous state is correlated with lifetime reproductive success (LRS, i.e., number of offspring produced during the entire lifespan of an individual[34]), longevity, litter size and juvenile survival in re-sequenced high coverage genomes of arctic foxes, (ii) investigate whether genome wide heterozygosity is correlated with the same fitness traits, (iii) compare the proportion of predicted deleterious mutations in re-sequenced genomes between native foxes and descendants of immigrants, and (iiii) compare fitness between native and immigrant descendants using a large, long term dataset on litter size, longevity and LRS.

## Results

The arctic fox population in Scandinavia is small and fragmented[35] and resides in the mountain tundra where it heavily relies on lemmings (*Lemmus lemmus*) and rodents as primary food source, which fluctuate in abundance in 3–5 year cycles. The subpopulation in this study is located in Helagsfjällen (3400 km²), Jämtland county (62°N, 12°E) and is the most southern and geographically isolated subpopulation in Sweden.

We performed whole-genome re-sequencing of 30 arctic foxes and mapped paired-end reads to a red fox (*Vulpes vulpes*) de novo assembly (PRJNA378561 [https://www.ncbi.nlm.nih.gov/bioproject/PRJNA378561/])[36] in order to avoid annotation bias. We obtained a

depth of coverage of 19x – 39x (table S1) with an average of 27x. Using an annotation (GCF_003160815.1 [https://www.ncbi.nlm.nih.gov/datasets/genome/GCF_003160815.1/])[36], we identified sites within coding regions carrying putatively deleterious loss of function (LoF) variants which have a disruptive impact on protein function. We also identified missense variants, which are non-disruptive variants that may affect protein effectiveness and synonymous variants which are unlikely to change protein behavior. In total, we found 1300 LoF variants, 44,433 missense variants and 86,508 synonymous variants in the population. A site frequency spectrum showed a downward shift in the frequency distribution of LoF mutations compared with missense and synonymous mutations, i.e., LoFs had a higher percentage of rare alleles whereas there was a higher percentage of fixed or nearly fixed missense and synonymous mutations (Fig. S1). We also calculated genome wide heterozygosity as the number of heterozygous sites per 1 kb based on mapping reads to an arctic fox assembly (PRJNA704825 [https://www.ncbi.nlm.nih.gov/bioproject/PRJNA704825/])[33].

### Genomic variation and fitness

Using data from a long term pedigree[32] and information about individual ancestry[37], we linked individual fitness traits to genomic variation. The metrics of genomic variation used were the proportion of LoF, missense, and synonymous genotypes in homozygote state. These metrics were used to investigate the effect of expressed mutations of different impact levels (i.e., expressed mutational load) on fitness. We also included genome wide heterozygosity in the analyses to investigate the effect on fitness across the entire genome. The fitness traits that we included were LRS, longevity, litter size, and juvenile survival. We also controlled for individual sex as well as which phase of the rodent cycle each individual was born during. To increase the statistical power, we used both individuals from Helagsfjällen, the main focus of this study as well as seven additional foxes from a more northwards located subpopulation in Sweden (Vindelfjällen-Arjeplogsfjällen), making a total of 37 individuals. Of the 37 re-sequenced foxes, the rodent abundance at birth was known for 35 individuals and they were hence included in the analyses on the effect of genomic variation on juvenile survival. Of these, 21 individuals survived their first year of life and these were thus included in the longevity analysis (age span: 1–9 years old). LRS and litter sizes could be confidently assigned for 18 individuals.

We found that a high proportion of LoF genotypes in homozygous state in adult foxes had a negative effect on LRS (linear model for square root of LRS, $n = 18$, $R^2 = 0.30$, $t = -2.24$, $p = 0.042$, 95% CI [−6506, −147], Fig. 1a; Table S2) and longevity (linear model for logarithm of age, $n = 21$, $R^2 = 0.36$, $t = -3.09$, $p = 0.007$, 95% CI [−2548, −482], Fig. 1b; Table S3). There was however no statistically significant effect of the proportion of homozygous LoFs on litter size (linear model, $n = 18$, $R^2 = 0.13$, $t = -1.34$, $p = 0.202$, 95% CI [−10885, 2517], Fig. 1c; Table S4), but a trend for higher probability of juvenile survival in individuals with lower proportion of homozygous LoFs (generalized linear model, $n = 35$, $z = -1.86$, $p = 0.062$, Fig. 1d; Table S5). We detected no effect of the proportion of heterozygous LoF mutations (i.e., masked mutational load) on fitness (LRS: $R^2 = 0.05$, $t = -0.14$, $p = 0.889$, 95% CI [−5656, 4952]; longevity: $R^2 = 0.009$, $t = 0.38$, $p = 0.707$, 95% CI [−1801, 2596]; litter size: $R^2 = 0.03$, $t = 0.23$, $p = 0.825$, 95% CI [−9106, 11238]; juvenile survival: $z = -1.05$, $p = 0.295$).

Moreover, we found that the probability of juvenile survival was negatively influenced by the proportions of homozygous missense ($z = -2.41$, $p = 0.016$; Fig. 1h; Table S9) and synonymous ($z = -2.41$, $p = 0.016$ Table S13) genotypes, and positively influenced by genome wide heterozygosity ($z = 2.57$, $p = 0.010$; Fig. 1l; Table S17). We detected no statistically significant effect of proportion of homozygous missense (Fig. 1e–g; Tables S6–8) or synonymous genotypes (Tables S10–12) or genome wide heterozygosity (Fig. 1i–k; Tables S14–16) on LRS, longevity or litter size. Individual sex and

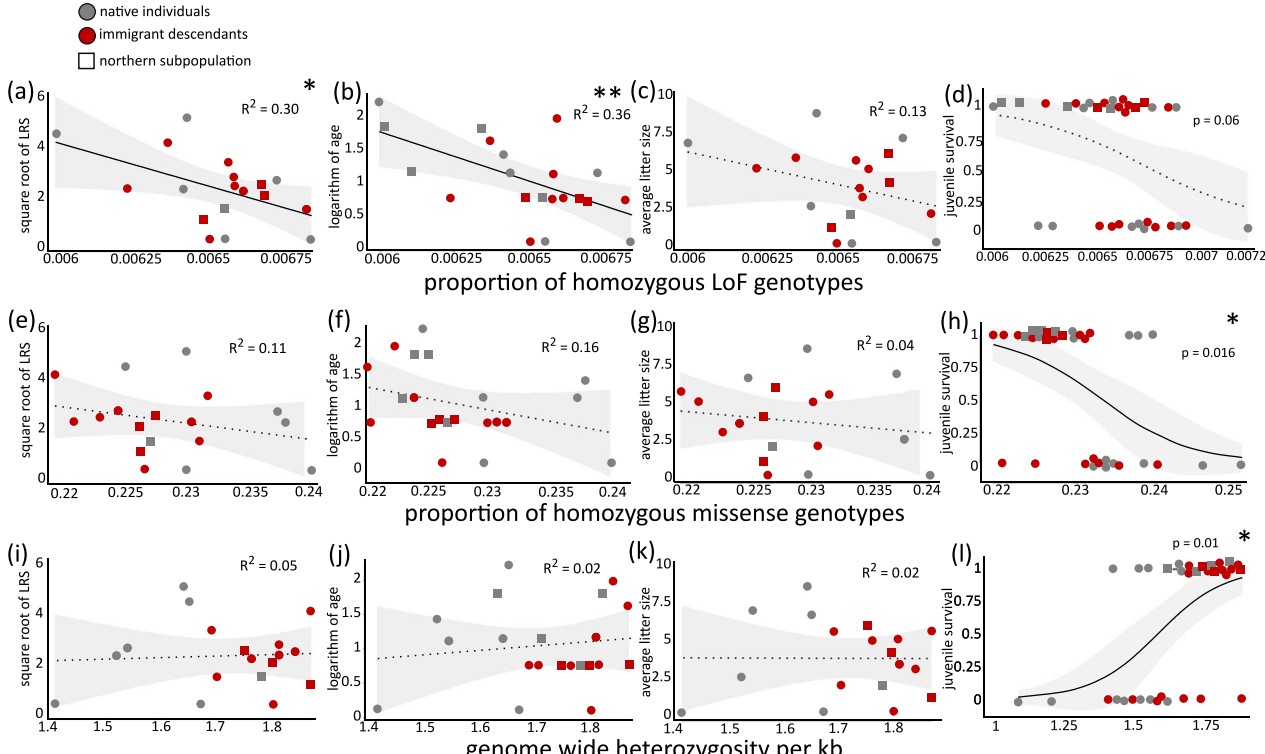

**Fig. 1 | Genomic variation and fitness in a Swedish arctic fox population.**
Relationship between proportion of homozygous loss of function (LoF) genotypes, lifetime reproductive success (LRS; $p = 0.042$; **a**), longevity ($p = 0.007$; **b**), litter size ($p = 0.202$; **c**) and juvenile survival ($p = 0.062$; **d**). Relationship between proportion of homozygous missense genotypes, LRS ($p = 0.319$; **e**), longevity ($p = 0.092$; **f**), litter size ($p = 0.586$; **g**) and juvenile survival ($p = 0.016$; **h**). Relationship between genome wide heterozygosity per kb, LRS ($p = 0.964$; **i**), longevity ($p = 0.592$; **j**), litter size ($p = 0.888$; **k**) and juvenile survival ($p = 0.010$; **l**). Gray data points represent native individuals and red ones represent immigrant descendants. Circles represent individuals from the focal subpopulation (Helagsfjällen) and squares represent individuals from a more northwards located subpopulation (Vindelfjällen-Arjeplogsfjällen). Solid trend lines and * mark statistically significant relationships (**$p < 0.01$, *$p < 0.05$) and dashed trend lines mark non-significant ones, two-sided linear models (**a–c**, **e–g**, **i–k**) and generalized linear models (**d**, **h**, **l**). $P$ values were not adjusted for multiple testing. Shaded areas indicate 95% confidence intervals. Source data are provided as a Source Data file.

whether foxes were born during increasing or decreasing rodent abundance had no statistically significant effect on fitness in any of the above analyses. Genome wide heterozygosity correlated strongly with the proportion of homozygous synonymous ($r = -0.92$, 95% CI [−0.96, −0.85], $t = -13.72$, $p = 1.20 \times 10^{-15}$; Fig. S2a) and missense ($r = -0.88$, 95% CI [−0.94, −0.78], $t = -11.12$, $p = 4.9 \times 10^{-13}$; Fig. S2b) genotypes, and more weakly but still statistically significantly with the proportion of homozygous LoF genotypes ($r = -0.36$, 95% CI [−0.61, −0.04], $t = -2.27$, $p = 0.029$; Fig. S2c).

## Mutational load and ancestry

From the previously published pedigree, we obtained information on individual ancestry, and 15 of the 30 re-sequenced foxes from Helagsfjällen were native individuals and 15 were hybrids between immigrants and natives (F1–F3 generations). For these 30 individuals, we linked the proportion of LoF, missense and synonymous alleles respectively (taking alleles in both homozygote and heterozygote state into account) to individual ancestry (native or hybrid) to investigate whether the immigration event potentially introduced new deleterious variation into the population. Despite previously demonstrated lower inbreeding and higher genome wide heterozygosity in immigrant F1 (native x immigrant)[33], we found that immigrant descendants (F1, F2 and F3 generations) had statistically significantly higher proportion of LoF alleles compared with natives (Mann–Whitney U-test, $p = 0.009$, 95% CI [0.0001, 0.0006]; Fig. S3a). Whereas native individuals had a larger variation in proportion of LoF alleles, all immigrant descendants had uniformly high mutational load (Fig. S3a). In the native gene pool we found a total of 1168 LoFs and in the hybrid gene pool we found

1220 LoFs, corresponding to 4.5 % higher number of LoFs in hybrids. Moreover, we found 132 LoFs in hybrids which were absent in the native gene pool. We found no statistically significant difference in the proportion of missense (Mann–Whitney U-test, $p = 0.217$, 95% CI [−0.0002, 0.0020]; Fig. S3b) or synonymous (Mann–Whitney U-test, $p = 0.068$, 95% CI [−0.0024, 0.00002]) mutations between native foxes and immigrant descendants.

## Gene ontology (GO) overrepresentation test on ancestry, litter size and longevity

We found 132 LoF variants in immigrant descendants that were completely absent in native foxes. Of these, 16 had a frequency of 20% or higher (20–37% in frequency; Table S18). Gene ontology (GO) analysis on these 16 gene variants indicated that various biological processes were associated with them, but none of them were overrepresented in the enrichment analysis when controlling for multiple testing.

We detected 30 LoF variants with higher frequency in individuals with the lowest fecundity (average litter size = 0–2 cubs, $n = 4$) compared with those with the highest fecundity (average litter size = 6.67–9.5 cubs, $n = 4$) using the criteria described in the methods section. At least half of the individuals that produced small litters were homozygous for eight of these variants (Table S18). According to the gene ontology analysis no biological process was however overrepresented.

We found only one gene variant (in ZNF503), to be homozygous in all individuals with short lifespan (1–3 years, $n = 10$) and heterozygous or absent in the individuals with long life spans (4–9 years, $n = 4$).

ZNF503 acts as a transcriptional repressor in vertebrates, critical for organ development[38]. However, 15 LoF variants were homozygous in at least 40% of the individuals with short life span (Table S18). Again, no process was overrepresented in the enrichment analysis of these. The fitness effects for the identified LoFs are unknown and it is likely that some of the identied LoFs have no or very little impact on fitness although they satisfy the above set criteria[39]. This will lower the likelihood of finding overrepresented processes.

### Ancestry and fitness

To study how fitness was shaped by individual ancestry, litters with known ancestry that had been produced since 2010 (when the first immigration event occurred) were used. For the analyses on LRS and longevity, foxes born during 2010–2015 with known ancestry that survived their first year of life were used ($n = 85$; Table 1). We performed linear mixed effect models controlling for sex, which phase of the rodent cycle each individual was born during (fixed effects) as well as their natal dens (random effect; Table S19–S20). Immigrant F2 and F3 individuals produced on average 5.25 (95% CI [1.45, 9.05]) fewer cubs (linear mixed effect model for square root of LRS: $t = -2.49$, $p = 0.013$; Fig. 2a; Table S19) and lived on average 1.6 (95% CI [0.1, 3.1])

years shorter lives (linear mixed effect model for logarithm of age: $t = -1.97$, $p = 0.049$; Fig. 2b; table S20) compared with immigrant F1. Native foxes did not differ from the other groups in LRS (immigrant F1: $t = 1.84$, $p = 0.07$, 95% CI [−0.09, 1.66]; immigrant F2 + F3: $t = -0.80$, $p = 0.42$, 95% CI [−0.85, 0.53]; Fig. 2a; Table S19) or longevity (immigrant F1: $t = 1.09$, $p = 0.27$, 95% CI [−0.28, 0.71]; immigrant F2 + F3: $t = -0.98$, $p = 0.33$, 95% CI [−0.61, 0.18]; Fig. 2b; Table S20). There was no difference in LRS ($t = -0.45$, $p = 0.66$, 95% CI [−0.44, 0.72]) or longevity ($t = 0.16$, $p = 0.87$, 95% CI [−0.28, 0.42]) between males and females and no difference in longevity between foxes born during increasing or decreasing rodent abundance ($t = -1.22$, $p = 0.22$, 95% CI [−0.49, 0.23]). However, individuals born during increasing rodent abundance produced on average 2.4 (95% CI [1.2, 3.6]) more cubs during their lifetime compared to those born during decreasing abundance ($t = 2.68$, $p = 0.007$; 95% CI [0.003, 1.16]).

When assessing ancestry and litter size, litters were grouped into four different pair combinations (Table 1; see Methods). To control for differences in territory quality and resource abundance between years, we used which phase of the rodent cycle each litter was produced during as fixed effect and natal den as random effect (Table S21). During 2010 to 2019, 166 litters were produced, of which 121 were of known ancestry and hence included in the litter size analysis (Table 1). We found that immigrant F1 litters (native x immigrant) were larger compared with the litters produced by all other pair combinations (linear mixed effect model, native x native: $t = 3.98$, $p = 6.85 \times 10^{-5}$, 95% CI [2.78, 6.96]; native x immigrant F1-F4: $t = 5.33$, $p = 9.82 \times 10^{-8}$, 95% CI [3.99, 8.07]; immigrant F1-F4 x immigrant F1-F4: $t = 3.75$, $p = 1.80 \times 10^{-4}$, 95% CI [2.86, 7.38]; Fig. 2c; Table S21) with an average of 11.17 cubs per litter (Table 1).

Further, purebred native pairs (native × native) produced on average 1.16 (95% CI [0.31, 2.01]) more cubs per litter compared with descendants of immigrants that backcrossed with natives (native x immigrant F1-F4; $t = 2.56$, $p = 0.010$; Fig. 2c). Litters that were backcrosses within immigrant lineages (immigrant F1-F4 x immigrant F1-F4) did not differ in size from native litters (native x native pairs; $t = 0.11$, $p = 0.91$, 95% CI [−1.08, 1.57]; Fig. 2c) but showed a trend for larger litters compared with native backcrosses (native x immigrant F1-F4 pairs; $t = 1.9$, $p = 0.057$, 95% CI [−0.18, 2.02]; Fig. 2c) although not statistically significant. Litter size also varied between phases of the rodent cycle with larger litters being produced during years with decreasing rodent abundance ($t = 3.27$, $p = 0.001$, 95% CI [0.91, 2.67]).

**Table 1 | Average litter size, life span and lifetime reproductive success (LRS) in different ancestry classes of a Swedish arctic fox population, as well as their respective sample sizes**

| Ancestry pair combination | Fitness traits litter size (n. cubs) | Sample size n. litters |
|---|---|---|
| native x native | 6.30 | 37 |
| native x immigrant | 11.17 | 6 |
| immigrant x immigrant F1-F4 | 5.14 | 59 |
| immigrant F1-F4 x immigrant F1-F4 | 6.05 | 19 |
| **individual ancestry** | **life span (yrs)**   **LRS (n. cubs)** | **n. individuals** |
| native | 3.8   7.60 | 30 |
| immigrant F1 | 4.8   12.19 | 21 |
| immigrant F2 + F3 | 3.2   6.94 | 34 |

Litters were divided into different pair combinations of native and immigrant descendants. Average life span and lifetime reproductive success were assessed for individual foxes of different ancestry that survived their first year of life.

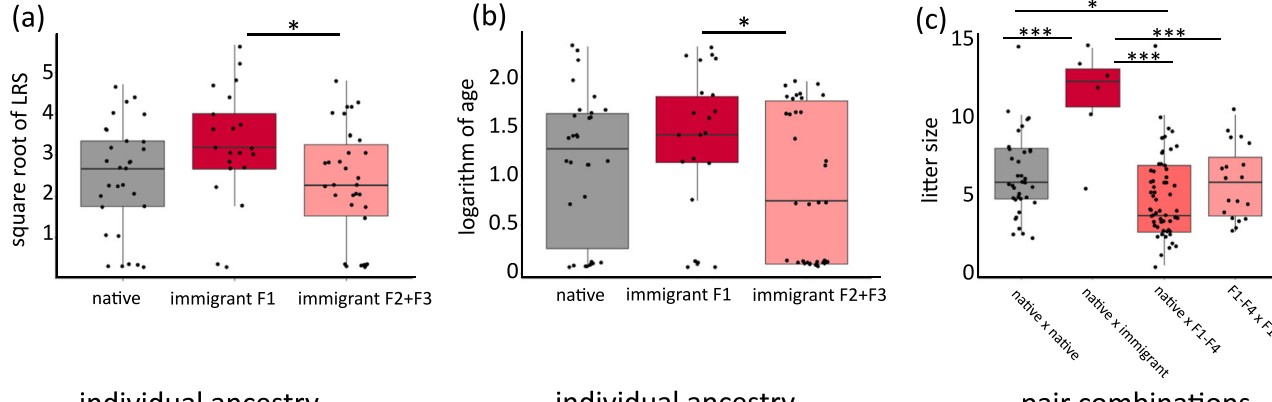

## Fig. 2 | Individual ancestry and fitness in a Swedish arctic fox population.

Lifetime reproductive success (LRS; **a**), longevity (**b**) and litter size (**c**) in native individuals and immigrant descendants. Lifetime reproductive success (**a**) and longevity (**b**) were analysed for individuals with known ancestry that survived to adulthood ($n = 85$). Litter size (**c**) was analysed based on four different native and immigrant pair combinations ($n = 121$). F1-F4 represent first to fourth generation

offspring of immigrants. * marks statistically significant results (two-sided linear mixed effect models). (***$p < 0.001$, *$p < 0.05$, exact $p$ values found in supplementary material Table S19–S21). $P$ values were not adjusted for multiple testing. Middle lines within boxplots represent medians, boxes represent the 25th–75th percentile. Vertical lines represent minima and maxima. Source data are provided as a Source Data file.

## Discussion

The theories of inbreeding depression and genetic rescue are corner stones within conservation genetics but the underlying mechanisms are still not well understood. A number of studies on wild populations have connected genome wide inbreeding to fitness[27,33,40–46] but empirical evidence linking deleterious mutations to fitness in natural populations remains rare. We here show that arctic foxes with a higher proportion of homozygous LoF genotypes have lower LRS and live shorter lives than those with lower proportions. The fitness effect was prevalent for homozygotes but not heterozygotes, which implies that the reduced fitness is caused by inbreeding depression through increased expression of recessive deleterious mutations, which has strong support in theory[47–50]. Moreover, genome wide heterozygosity correlated with the proportion of synonymous, missense and LoF genotypes in homozygote state, reflecting that increased heterozygosity across the genome reduces the homozygosity within coding regions and lowers the expression of putative recessive deleterious mutations.

We also demonstrate that the probability of juvenile survival is negatively associated with the proportion of homozygous missense and synonymous genotypes and positively associated with genome wide heterozygosity. We thus conclude that deleterious mutations of both large effect and those of smaller effect are important drivers of inbreeding depression in this Scandinavian arctic fox population. As the effect appears to be trait specific, different genomic mechanisms are likely involved in traits related with reproduction and survival. Similarly, in a population of hihi (*Notiomystis cincta*), juvenile survival was found to be affected by genome wide inbreeding, whereas LRS could be associated with a small number of SNPs[28]. Our results provide one of the first direct evidence of a link between putatively deleterious mutations and fitness in a wild endangered animal population. Strongly deleterious alleles have higher potential to be purged, since they are often exposed to strong selection pressure[50]. Our site frequency spectrum revealed a downward shift in the frequency of LoF mutations compared with missense and synonymous mutations, implying that purifying selection has acted on strongly deleterious alleles. However, as juvenile survival is associated with the proportion of missense genotypes, which are assumed to be more mildly deleterious, inbreeding depression is likely to persist despite purging of highly deleterious mutations. Moreover, it was recently demonstrated that purifying selection can act on deleterious alleles but still fail to effectively reduce inbreeding depression, as other deleterious alleles with inbreeding-associated fitness instead rise in frequency[29,51], which may be the case in our study population as well. As the population is extremely small there is an imminent risk that genetic drift has a stronger impact on allele frequency change than purifying selection[52], which could lead to fixation of a subset of LoFs.

Due to the fluctuating food resources, arctic foxes in Scandinavia are highly dependent on producing large litters during rodent increase and peak years. This is an important life history trait that enables the population to recover in size after years of low food abundance[53]. Our finding that expressed mutational load leads to lower LRS suggests that inbreeding depression could have a negative impact on the population recovery, which may be further enhanced in combination with the resource fluctuations. Moreover, the negative relationship between longevity and expressed mutational load is also worrying, as arctic foxes reproduce throughout their lives[54]. With a lemming cycle of approximately four years, it is important that individuals reach that age to utilize both the years of increasing and peaking rodent abundance. Specific LoF variants associated with fitness were involved in many different biological processes but no processes were over-represented. A polygenic inheritance pattern has previously been found to influence both litter or clutch size[55–57] and longevity[58,59] in other species in the wild. Moreover, the strong effect of environmental factors, in particular rodent abundance, on arctic fox fitness[60]

complicates the identification of specific mutations involved in reproductive output and longevity.

The risk and impact of introducing deleterious alleles through gene flow has recently been debated within conservation genetics[8–10]. In this study, the gene pool of immigrant descendants had higher mutational load compared with native individuals, although the effect size of 4.5% LoF alleles was small. This indicates that the immigrants have brought in a small number of new deleterious variants into the population, which were previously absent (e.g., through earlier genetic drift or purging). However, genomes of immigrants need to be sequenced to investigate in more detail the genomic variation introduced. We also found that immigrant F1 litters were larger compared with purebred native litters. This, in combination with previously documented elevated fitness in immigrant first generation offspring[31] can be explained by the masking of recessive deleterious alleles. However, in subsequent immigrant generations the elevated litter size was reduced. In fact, immigrant descendants that backcrossed with natives produced even smaller litters compared with purebred native litters. Further, immigrant F2 and F3 individuals lived shorter lives and had lower LRS compared with immigrant F1 individuals. This could be attributed to increased expression of recessive deleterious alleles originating from both native and immigrant lineages. Furthermore, in later years the population size has decreased in comparison with the first years following the immigration events, in combination with increasing inbreeding levels[32].

Population genetic theory predicts that large outbred populations should harbor a high level of segregating recessive deleterious alleles compared with populations with long term small effective population size[61,62], which has also been observed in a range of species[46,63–68]. However, historically small populations are more likely to have a higher degree of fixed deleterious alleles and hence lower fitness[69] and are often in urgent need of gene flow. Simulations have suggested that recipient populations that are very small may not be able to purge deleterious variation introduced by migrants, but instead risk being subject to increased inbreeding depression and extinction[70]. The three arctic fox immigrants originated from a captive breeding program that breeds foxes from different subpopulations throughout Scandinavia with each other[71]. For instance, two of the immigrants that established in our study population were a genetic mix of five different subpopulations from mid to northern Norway[33]. Although this has likely resulted in high genome-wide variation, it could also have accumulated a high number of recessive deleterious alleles in these individuals. The arctic foxes in Scandinavia consisted of one large continuous population less than 150 years ago[72,73]. However, many of the fragmented subpopulations are now extremely small[74] which could have led to accumulation of different deleterious mutations in each subpopulation due to strong genetic drift. The mutational load of the admixed foxes from the breeding program might thus be comparable with individuals originating from large and outbred populations. Instead, we hypothesize that translocating individuals between closely located subpopulations could lead to less inbreeding depression rather than mixing individuals from populations further apart. We propose that future analyses should compare the deleterious variation between subpopulations as well as include genomes of immigrants. Identification of deleterious variation associated with fitness in populations in need of gene flow provides a powerful tool to maximize the amount of genetic diversity during translocation while at the same time minimizing the effect of introduced deleterious variation.

## Methods

The research compiles with all relevant ethics regulations, including permits to work in protected habitats, approved by the county board administration of Jämtland (521-4797-2017, 521-2593-2017) and Västerbotten (521-3191-2014, 521-4640-2019). Permits to catch, handle and eartag foxes were approved within the ethical permit from the Swedish

Board of Agriculture (A49-01, A36-11, A130-07, A18-14, A10-17, A130-07) and additional allowances from Swedish Environmental Protection Agency (412-4191-03 Nf, 412-5362-04 Nf, 412-7884-07, NV-01959-14, 412-35-99 Nf, NV-02547-17).

## Study population
The subpopulation in Helagsfjällen, which is the main focus of this study, went functionally extinct in the 1980's but was re-founded in 2001[30] and has been closely monitored since. Until 2010 the population was founded by only seven individuals (of which six are genetically represented in the current population)[32]. Additional immigration was documented in 2010 and 2011, when three male foxes established and reproduced in the population. The immigrants originated from a captive breeding station in Norway and had been released in a Norwegian subpopulation located approximately 120 km from the study population[31]. Two of the immigrants were brothers and originated from five subpopulations from central to northern Norway. The third immigrant was an unrelated male and originated from two subpopulations in northern Norway[31]. The immigrant brothers were more successful than the unrelated one. In 2019, their respective ancestry could be traced to one third of all litters produced, whereas the ancestry of the unrelated male was traced to 17% of the litters[32]. Although the population has increased in size since the founding event in 2001, it has remained very small during the whole study period and the number of reproducing adults have fluctuated between 0 and 60 individuals in response to the rodent cycle[32].

## Fitness traits
Every summer the population is monitored by documenting number of litters, litter sizes and identifying surviving adults at all known den sites. The cubs are captured and ear tagged with individual color combinations, which enables visual identification of individual foxes during following years without the need to re-trap them. In this study, we used the fitness traits LRS, longevity, litter size and juvenile survival, verified through a previously published pedigree[32]. Life history of the Scandinavian arctic fox is heavily influenced by the rodent cycle, which is characterized by a peak in rodent abundance every third to fourth year, generally followed by a crash and an increase phase[75]. Most arctic foxes die already as cubs, especially those born during years of decreasing food abundance[76]. Foxes that survive to adulthood rarely live through more than one rodent cycle (i.e., 3–4 years)[77]. The oldest fox visually observed in the study population however became at least eleven years old. Arctic foxes become sexually mature already as one year olds, and age at first reproduction generally varies between one and four years of age[78]. Cubs are born around May and litter sizes vary from one up to 18 cubs depending on the rodent abundance, with an average litter size of 6.6 cubs[53,78].

In this study, we assessed litter size by counting the maximum number of cubs visually observed in each litter. All dens were visited at least once during July, the time that cubs usually emerge up on the dens. During the visits, the dens with cubs were monitored for a minimum of 24 h[79]. Juvenile survival was determined as whether or not individuals survived their first year of life (i.e., 1 or 0) and was based on both visual observations and genetic parentage assignment. Longevity was estimated as years of life and we assumed that individuals that had not been sighted for a full rodent cycle (i.e., at least three years) died after the last observation[31]. LRS was estimated as the total number of offspring an individual produced during its entire lifespan[34].

## Sample selection and re-sequencing
We performed whole genome sequencing on 37 individuals that had been ear tagged between 2001 and 2019, of which 30 were previously published (PRJEB43377 [https://www.ncbi.nlm.nih.gov/bioproject/?term=PRJEB43377] and PRJEB55788 [https://www.ebi.ac.uk/ena/browser/view/PRJEB55788])[33,80]. The samples were chosen based on a genetically verified pedigree[32] and information on individual ancestry[37] to represent both native and immigrant descendants as well as to obtain variation in fitness. In total we sampled 19 native foxes and 18 immigrant descendants, ranging from immigrant F1 (native x immigrant) to immigrant F4 generations. In total, 30 of the re-sequenced foxes originated from the subpopulation of Helagsfjällen, whereas seven foxes originated from a subpopulation located more northwards in Sweden (Vindelfjällen-Arjeplogsfjällen)[37] and share the same demographic history.

During ear tagging we obtain a small piece of tissue for each individual which were used for DNA-extraction. For the 23 previously published samples from Helagsfjällen (PRJEB43377 [https://www.ncbi.nlm.nih.gov/bioproject/?term=PRJEB43377]), DNA was extracted and sequenced as described in Hasselgren et al.[33]. For the remaining samples, the Qiagen DNeasy Blood & Tissue kit (cat.no. 69504) was used for DNA extraction. DNA-concentration was quantified using the Qubit dsDNA Quantification High Sensitivity (cat.no. Q32851) and Broad Range (cat.no. Q32850) assay kits from ThermoFisher. To increase the DNA-concentration, Vivaspin centrifugal concentrators (cat.no. Z614041) from Sigma-Aldrich were used. The samples were sequenced at the National Genomics Infrastructure (SciLifeLab) in Stockholm using the Illumina NovaSeq6000 and MiSeq platforms with a 2 × 150 bp setup and TruSeq PCR-free library construction in accordance with the general procedures at SciLifeLab.

## Bioinformatics and variant calling
Reads were processed in multiple steps following Kutschera et al.[81]. First, adapters were trimmed from raw reads using Trimmomatic version 0.36[82]. Trimmed reads were mapped to a red fox de novo assembly (PRJNA378561 [https://www.ncbi.nlm.nih.gov/bioproject/PRJNA378561/])[36] using the Burrows-Wheeler Aligner (BWA) version 0.7.17[83]. BAM-files were indexed and sorted using Samtools version 1.8[84], duplicates were removed using Picard version 2.10.3 (https://broadinstitute.github.io/picard/) and indels were realigned using GATK version 3.7 (https://software.broadinstitute.org/gatk/). We used Qualimap version 2.2 for quality control and to obtain mean depth of coverage[85]. Variant calling was performed using bcftools mpileup version 1.8[84,86] using a minimum depth of coverage of one third of the average coverage and a maximum of twice of the average coverage. We used the same software to filter out sites with a quality score lower than 30 and SNPs within 5 bp of indels. Finally, we filtered out hard masked repeat regions using BEDTools version 2.27.1[87], merged all samples into a single vcf-file and obtained a total of 16 703 773 high quality SNPs. For downstream analyses we only kept variants called in all individuals, which made up a total of 10 880 359 SNPs.

## Mutational load
We annotated synonymous and non-synonymous variants within coding regions by using the red fox annotation (GCF_003160815.1 [https://www.ncbi.nlm.nih.gov/datasets/genome/GCF_003160815.1/])[36]. We used SNPeff version 4.3[88] to generate a database for red fox, using the protein sequences extracted from its annotation and obtained a total of 37,937 genes. We removed in-frame stop codons from the annotation using the -V option in cufflinks version 2.2.1[89,90] and obtained 19,217 genes. Second, we identified three categories of impact on protein expression, according to the SNPeff manual[88]: (i) low: variants that are unlikely to change protein behavior, hereafter referred to as synonymous (ii) moderate: non-disruptive variants that might change protein effectiveness, hereafter referred to as missense and (iii) high: variants assumed to have high disruptive impact in the protein, probably causing protein truncation, LoF or triggering nonsense mediated decay (e.g., stop codons), hereafter referred to as LoF variants. All LoFs were checked manually to safeguard against sequencing errors by comparing the gene sequences of the red fox assembly with the vcf-file of each individual.

Finally, we calculated the proportion of derived alleles in each impact category (LoF, missense and synonymous alleles) for every individual to ensure that counts were not biased because of sequencing completeness. This was done by dividing the total number of alleles in respective category by the total number of scored alleles (similarly as e.g., Valluru et al.[91]). For example, to calculate the proportion of LoF alleles we divided the number of LoF alleles by the number of LoF + missense + synonymous alleles for each respective individual. We used the same approach to calculate the proportion of genotypes in homozygous (expressed mutational load) versus heterozygous state (masked mutational load) for each impact category. Here we divided the number of genotypes in an individual (for example the number of homozygous genotypes for LoF alleles in a particular individual) by the total number of scored genotypes (LoF + missense + synonymous genotypes). We removed variants that were fixed across all individuals.

### Gene ontology (GO) overrepresentation test
We searched for biological functions of LoF variants that were absent in native foxes and with a frequency of 20 % or higher in immigrant descendants in the Panther gene ontology (GO) classification system using the domestic dog (*Canis lupus familiaris*) as reference. We also identified biological functions of LoF variants that differed in frequency between individuals with small versus big litters or long versus short lives. To identify candidate variants underlying variation in litter size, we compared LoF variants in the four foxes with the lowest fecundity (average litter size = 0–2 cubs) with the four foxes with the highest fecundity (average litter size = 6.67–9.5 cubs). We reasoned that since litter size can be affected both by the number of recessive deleterious mutations in homozygous state in the mother but also through inheritance of recessive deleterious alleles from two heterozygous parents, we set the following criteria to identify such damaging variants: variants must (i) be homozygous or heterozygous in the individuals with small litters, but cannot be absent, and (ii) either be absent or heterozygous in the individuals with large litters. To identify LoF variants potentially affecting longevity, we set the following criteria: variants must be (i) heterozygous or absent in the individuals with the longest lifespans (4–9 years, $n = 4$) and (ii) in homozygous state in individuals with the shortest lifespans (1–3 years, $n = 10$). We tested for statistical overrepresentation of biological functions in the genes identified using Fisher's exact test with Bonferroni correction for multiple testing.

### Genome wide heterozygosity
Genome wide heterozygosity was quantified from the mapping to an arctic fox reference genome (PRJNA704825 [https://www.ncbi.nlm.nih.gov/bioproject/PRJNA704825/])[33]. We estimated genome wide heterozygosity as the number of heterozygote sites per 1 kb with the population mutation rate ($\theta$) which approximates the expected heterozygosity per site under infinite sites model using mlRho version 2.9[92]. Bases with quality below 30, reads with mapping quality below 30, and positions with root-mean-squared mapping quality below 30 were filtered out. Since high and low coverage in certain regions, due to structural variation, can result in inaccurate mapping and false heterozygous sites, we filtered out sites with depth below one third of the average coverage and twice the average coverage for each sample.

### Statistical analyses
We performed non-parametric Mann-Whitney U-tests to compare the proportion of deleterious alleles in the re-sequenced native foxes versus those with immigrant ancestry. We fitted linear models to estimate the effect of genome wide heterozygosity, the proportion of LoF, missense, and synonymous genotypes on LRS, longevity, and average litter size in the re-sequenced foxes that survived to adulthood. We fitted generalized linear models with binomial distribution to investigate how the proportion of LoFs, missense and synonymous genotypes, and genome wide heterozygosity influenced the probability of surviving the first year of life. Since LRS and longevity were not normally distributed, box cox tests were performed to give information about appropriate transformation of data. In accordance to these test, longevity was log-transformed and LRS was square rooted for all fitness analyses. We also controlled for the sex of each individual as well as which phase of the rodent cycle they were born during. To maintain statistical power, rodent cycles were merged into the phases "increase", and "decrease", determined by whether individuals were born during increasing or decreasing rodent densities[30–32]. Although individual ancestry has previously been documented to impact fitness[31,32] we decided to not include ancestry as a parameter in the statistical analyses based on re-sequenced genomes in order to maintain statistical power in this small dataset.

To find out whether immigrant ancestry influenced litter size, we used all litters with known ancestry that were produced during 2010 to 2019 based on a previously published pedigree[32] and grouped them into the following four different pair combinations: native x native, native x immigrant, native x immigrant F1-F4 (i.e., immigrant descendants that backcrossed with natives) as well as immigrant F1-F4 x immigrant F1-F4 (i.e., immigrant descendants that backcrossed within the same immigrant lineage). For the longevity analysis, we combined pedigree data[32] with visual observations of ear tagged foxes with known ancestry born 2010–2015 that survived their first year of life. The traits longevity and LRS were compared between native, immigrant F1 and immigrant F2 + F3 individuals. Immigrant F2 and F3 individuals were grouped together not to lose statistical power, since very few F3 individuals had been recorded to reach adulthood. To test if there was an effect of ancestry on litter size, longevity or LRS, we performed three linear mixed effect models using phase of the rodent cycle as fixed effect and natal den as random effect. For the longevity and LRS analyses, the sex of each fox was also included as a fixed effect. Phases of the rodent cycle were merged into increasing or decreasing rodent abundance to maintain statistical power[30–32]. All statistical tests were performed using R version 4.4.0.

### Reporting summary
Further information on research design is available in the Nature Portfolio Reporting Summary linked to this article.

## Data availability
The re-sequencing data generated in this study have been deposited in the European Nucleotide Archive (ENA) under the accession numbers: PRJEB76449, PRJEB43377 and PRJEB55788. The datasets generated in this study are available in the Dryad database: https://doi.org/10.5061/dryad.7wm37pw2r. Source data are provided with this paper.

## Code availability
The r-code generated in this study are available in the Dryad database: https://doi.org/10.5061/dryad.7wm37pw2r.

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

## Acknowledgements

K.N. acknowledges support from the Swedish Research Council FORMAS (no. 2015-1526 and 2020-01402). A.A. acknowledges support from the EU-life projects SEFALO and SEFALO + . The study was also financed

by EU/Interreg Sweden-Norway to Felles Fjellrev I (no. 304-4159-13) and II (no. 20200939). We acknowledge support from Science for Life Laboratory, the Knut and Alice Wallenberg Foundation, the National Genomics Infrastructure funded by the Swedish Research Council, and Uppsala Multidisciplinary Centre for Advanced Computational Science for assistance with massively parallel sequencing and access to the UPPMAX computational infrastructure. We thank Verena Kutschera and Martin Kierczak for assistance with bioinformatics through the WABI Long-Term Bioinformatics Support, and to rangers and volunteers for conducting fieldwork. The computations were performed on resources provided by SNIC through Uppsala Multidisciplinary Center for Advanced Computational Science (UPPMAX) under Project SNIC 2020/15-131 and 2021/22-406.

## Author contributions

K.N., L.D., and M.H. designed the study. M.H., K.N., and A.A. collected the data. M.H. and J.V.S. carried out the laboratory work. M.H. and N.D. analyzed the data. M.H. drafted the manuscript. All authors read and edited the manuscript.

## Funding

## Competing interests

The authors declare no competing interests.
