## [Peer Review File · Nature Communications]

Strongly deleterious mutations influence reproductive output and longevity in an endangered populationREVIEWER COMMENTS

Reviewer #1 (Remarks to the Author):

The objectives of this interesting study were to evaluate the relationship between fitness and homozygosity for putatively deleterious alleles, and to determine how ancestry affected the genetic load. The topic is very interesting and important for understanding the genetic basis of inbreeding depression and of how we can use genomic data to understand fitness in small, wild populations of conservation concern. The field data are clearly hard-won and of very high quality and the sequencing and bioinformatic analyses are well done.

There are several issues with the paper, some of which undermine the conclusions:

The finding of reinstatement of inbreeding depression upon backcrossing in either direction is quite interesting. While fitness varies among backcross categories (Figure 2), my general interpretation of the demographic data (at least in part shared by the authors in lines 300-301) is that both natives and immigrants seem to carry similarly substantial loads of deleterious recessive alleles. However, the paper's main focus related to genetic load brought by immigrants is on the genomic data, instead of on the demographic data which actually provide a direct measure of these effects. In particular, there's a big focus on the data shown in Figure S1a (e.g., 2nd to last sentence in the abstract and elsewhere in the results and discussion), which shows that immigrants have a very slightly higher value in some metric of genetic load associated with LOF alleles. The first issue with this is that the effect size, while statistically significant, seems miniscule in magnitude and thus unlikely to be biologically important. The second issue is that I cannot tell how this analysis was actually done. I provide more details on this issue below.

45: You could cite Darwin here, who was keenly interested in inbreeding depression in plants and in red deer estates.

47-49, 68-76: The study is set up by suggesting that it is not well understood whether large-effect loci often contribute to inbreeding depression. While the specific distribution of effects is not well understood, it is indeed clear that large effect, deleterious recessive alleles often play a crucial role. I suggest revising the introduction accordingly. There are very good data on this question in pedigree-based analyses of inbreeding depression in domesticated species, and in mice. Those analyses find that inbreeding depression is often largely due to identity-by-descent arising from a small subset of pedigree founders. If inbreeding depression was purely polygenic (i.e., lots of small-effect loci), then we would expect the contribution of different founders to inbreeding depression to be approximately even. The results from those studies suggest that loci with very large effects (probably a small number of lethal ($s=1$) or semi-lethal alleles (s 'near' 1)) comprise an important fraction of inbreeding depression. We reviewed some of these in our paper that is cited in this section (Kardos et al). I suggest considering those results and citing some of them. A few good citations are listed below, but there are other livestock and *Drosophila* studies that are highly relevant to this section, and also to the statement in lines 74-76.

Lacy et al. (1996, *Evolution* 50: 2187-2200)

Casellas et al. (2009, *J Anim Sciences* 87: 72-79)

Todd et al. (2018, *Scientific Reports* 8: 6167)

With respect to the Isle Royale example focused on strongly here, it is important to consider what is defined as as a locus with a large fitness effect ("strongly deleterious" in their paper) compared to large-effect loci that contribute to inbreeding depression in real populations. Robinson et al. defined a strongly deleterious locus as one with $Nes > 100$ (N_e = effective population size and s = the selection coefficient), where N_e was assumed to be 45,000 (left column of their page 7). This means that a 'strongly deleterious' allele had a selection coefficient of at least $s = 100/45,000 = 0.0022$ (~0.2%

increase in mortality for a derived allele homozygote compared to a wild type homozygote). I don't think this is a very useful definition of a large-effect locus for the purposes of this study for two reasons. First, alleles with such small effects ($s \leq 0.002$ or smaller) likely contribute little to inbreeding depression because they are more likely to have nearly additive effects (see citations in Kardos et al. 2021 PNAS). Second, lethal ($s = 1$) and semi-lethal alleles (s of say ~ 0.5 or larger), i.e. loci with truly very large fitness effects, are commonly observed in real populations and contribute substantially to inbreeding depression where they occur (i.e., highly deleterious alleles are thought to almost always be nearly completely recessive). Two direct observations of lethal recessives in birds are in Trask et al. (2016, *Journal of Animal Ecology*, 85, 879–891) and Ralls et al. (2000, *Animal Conservation*, 3, 145–153.). Consider focusing on empirical examples of the contribution of truly large-effect alleles from the literature that are based on direct demographic inferences (e.g., pedigree-based and direct analyses cited above).

54-66: An unmentioned important aspect of this issue is that when there is inbreeding depression, inbred immigrants have to be less likely to survive and reproduce in a new environment than immigrants with substantially higher heterozygosity. There are other benefits to using immigrants with high heterozygosity, including maximizing adaptive potential (see Ralls et al. 2020).

Section beginning at 126. There are several issues that I suggest carefully considering here:

The analysis presented here is meant to test whether homozygosity for LOF alleles (expected to have big fitness effects) is a better predictor of fitness than homozygosity for either missense and nonsynonymous alleles (expected to have small fitness effects, on average). The authors find that fitness is statistically significantly correlated with homozygosity for LOF alleles but not with homozygosity for missense or synonymous alleles, and thus conclude that inbreeding depression is largely driven by large-effect (LOF) alleles (the central finding of the paper: lines 20-25). I think the evidence is presently weaker than suggested in the paper.

There are some important and somewhat complex statistical problems with this analysis. First, the analysis does not account for other variables that are apparently also important (455-458) including temporal or spatial environmental variation and sex, which appear to be accounted for below in the separate analysis of the effects of ancestry on fitness (469-473). Why are these variables accounted for in the analysis of ancestry but not in the analysis testing for effects of homozygosity? Are the environment and sex constant across individuals used in the correlation analysis? That seems unlikely to be so because years of birth vary substantially across the sampled individuals (2001-2018, line 382), but nothing is mentioned in the paper about the distribution of sex among the individuals in the analysis. Assuming sex and environment (e.g., year of birth, natal birth place) vary among individuals in the analysis, then it seems they should be accounted for statistically. The sample size may be too small for the analysis to be very informative.

Second, it would be unsurprising if the difference in the observed correlations between fitness components versus homozygosity measured with different subsets of loci (LOF versus missense versus nonsynonymous loci) is due solely to a necessarily high sampling variance in the estimation of the correlation coefficients. With respect just to statistical phenomena, both p-values and regression coefficients are highly stochastic when sample sizes are so small. When inbreeding depression is present, we expect fitness (w) to be negatively correlated (on average) with homozygosity (H) measured at any set of polymorphic loci in the genome. This is expected even when we do not directly measure homozygosity at the loci that cause inbreeding depression, because of genome-wide identity disequilibrium that occurs when inbreeding varies among analyzed individuals. See several relevant papers from Patrice David, especially Szulkin et al. (2010, *Evolution*) which explains in detail an analogous statistical problem of testing for local effects driving heterozygosity-fitness correlations. The correlation coefficient (r) from a regression of w on H will vary across different sets of loci just due to sampling variation. Holding the strength of inbreeding depression constant, the sampling variance in r is expected to be higher when fewer loci are genotyped, minor allele frequency is lower (as is the case

with LOF loci), and when fewer individuals are analyzed (sample sizes here are extremely small). All three of these components of the sampling variance (sample size, number of loci, minor allele frequency) are important here, but do not appear to be accounted for statistically when evaluating the observed differences among models. The relevant question is not whether r varies among models that use different subsets of loci to measure H (they always do), or whether one is statistically significant and the other isn't, but rather whether the regressions vary more than expected by chance. I suggest modifying the statistical analysis to test whether the stronger correlation observed when H is measured with LOF variants compared to missense or nonsynonymous variants (the key finding of the paper, assuming this holds when other important variables are accounted for) is likely to be due to simple sampling variation in r among different sets of loci analyzed. I believe an F-ratio test could be used for this (see a similar application in Szulkin et al. (2010) from the pre-genomics era), but it seems that would only account for sampling variance arising from small sample size and would ignore effects of number of loci and allele frequency distributions. It might be possible to devise a permutation test, where the null distribution of the difference in r between models is based on permuted loci, which would also account for sampling variance arising from number of loci and allele frequency differences. Again the extremely small sample sizes are a major limitation. Lastly, restricting this analysis to include only coding loci with LOF, missense, and synonymous mutations means that one of the most important comparisons is not made. The coding part of genomes represents a small fraction of the loci that are subjected to purifying selection (e.g., <https://doi.org/10.1371/journal.pgen.1004525>). I therefore suggest comparing the association of fitness with homozygosity for LOF/missense/synonymous alleles versus homozygosity across the whole genome (i.e., homozygosity at all nearly 11,000,000 SNPs, line 414) which accounts for deleterious recessive effects across the entire genome.

163-167: Are the relevant methods for this in lines 428-430? It's unclear exactly what was done so I cannot interpret this result. What I think is the relevant method description is extremely short and says:

"We calculated the proportion of variants belonging to each category by dividing the number of alleles in respective category with the total number of alleles (LoF + moderate + synonymous). We also took genotypes in heterozygous and homozygous state into account."

This first refers to 'proportion of variants' which is the same thing as proportion of loci, but then says this is done by counting alleles, which is not the same as counting loci. I suggest defining this explicitly with an equation and clearly defined terms. Additionally, I cannot tell what is meant by "We took genotypes in heterozygous and homozygous state into account". It might be clearer to address topic in the paper using the standard R_x/y method of Do et al. (Nature Genetics 47, 126-131 (2015)), which would work well if the efficiency of historical purifying selection in one population versus the other is the focus of interest here. I cannot tell how it was calculated that "immigrant descendants had 4 % higher proportion of LoF alleles compared with natives" given the very small difference that is clear in figure S1A: ~ 0.0088 (native median) compared to ~ 0.0090 (immigrant descendants). It is hard to see this very small difference as being biologically important, but it is a major focus of the paper (see 2nd to last sentence in the abstract), but then again, I don't really understand how the analysis was done so it's impossible for me to interpret it properly.

240-242: There are many more examples from genetic rescue studies.

250: An insurmountable problem for analyses like this is that we do not know the distribution of the effects (selection coefficient) for any mutation class.

252-254: The statistical evidence for this finding is insufficient in my opinion, as described above in detail.

256: And consistent with lots of previous empirical data as noted above.

289: I suggest defining 'common' here. "16 LoF alleles that were common in immigrant descendants but absent in natives" represents about 1.5% of the total of 1,052 detected LOF alleles. It's a bit difficult to interpret this is being highly biologically important.

301-303: Is this based on the observation of "16 LoF alleles that were common in immigrant descendants but absent in natives"?

455-458: Please clarify whether the predictor for each individual is calculated as the proportion of genotyped LOF/missense/synonymous loci where the individual is homozygous for the DERIVED allele. This is an important detail that appears to be missing from them manuscript.

I also suggest that it would be useful to include a figure somewhere comparing the derived allele frequencies of derived LOF/missense/synonymous/ alleles (i.e., the sfs), which would help readers to understand why homozygosity is so much lower for LOF alleles than for missense alleles (x-axes in upper panels of Figure 1). If this has been done in a previous paper, it would be sufficient to state that the SFS are different and refer to the previous study.

I hope the authors find my comments helpful!
Marty Kardos

Reviewer #2 (Remarks to the Author):

Strongly deleterious mutations influence reproductive output and longevity in an endangered population

This manuscript addresses an important question in conservation biology: what role do deleterious mutations play in generating fitness variation, which - in turn - affects the long-term survival of endangered species? This question is addressed using empirical data from a small population of the endangered Arctic fox in Sweden, which, despite its geographical isolation does experience occasional immigration.

The main findings are that foxes with a high proportion of loss-of-function mutations live less long and produce fewer offspring, and that offspring of immigrants had on average 4% more of these loss-of-function variants. The latter finding suggests that immigrants brought in deleterious mutations into this endangered fox population.

While this work asks very interesting and important questions, it is difficult to know what to make of the results for a number of reasons. I will divide these reasons into biological and into statistical reasons.

First the biological reasons:

1) The major difficulty with this data set is the recent demographic history of this Arctic fox population. From 2001 to 2011 the population size increase from ~5 breeding adults to ~50, then crashing to ~5 breeders in 2016 before returning to ~30-40 in 2017-2019. In parallel, the average inbreeding in the population increased by a factor of ~10. Following the immigration of 3 males in 2010-2011, average inbreeding declined, although not as much as one might have expected because there was extensive inbreeding within the immigrant lineages, including parent-offspring matings (all based on Lotsander et al. 2021). Thus, effective population size, average inbreeding, proportion of immigrant ancestry, phase of the lemming cycle, and probably several other factors related to fitness (such as those reported in previous publications by the authors, e.g. Erlandsson et al. 2017 or Choi et al. 2019) all covary systematically with each other. This makes it extremely challenging to quantify the contribution of deleterious mutations to fitness variation, because a given deleterious mutation is likely

present in homozygous state only during one particular phase of the lemming cycle in one or a few individuals with certain ancestries. Thus, separating environmental effects from genetic effects seems high impossible, especially with a sample size of 16 and 14, respectively. At the very least, the confounding of inbreeding, immigrant ancestry, and environmental factors would need to be explored in much more detail in the manuscript or the supplementary material before one can have confidence in the results.

2) Unfortunately, one of the main genomic measures used, the proportion of homozygote genotypes of a particular class, is never defined explicitly. I assume it equates to the number of homozygotes of a particular class (e.g. loss-of-function) divided by the total number of variable sites of this class. If this is correct, then the numbers on the x axes in Figure 1 suggest that for LoF variants the range of number of homozygous mutations between the two most extreme individuals is $0.005 \times 1052 = 5.26$ and $0.007 \times 1052 = 7.36$, and for missense variants 9330 and 10664, respectively. Would this not argue for a far more important role of the mildly deleterious missense mutations than the LoF mutations? A similar reasoning would suggest that a regression of fitness traits on the proportion of homozygotes would be the more appropriate measure of mutational load than the unitless correlation. I don't think a unitless measure can serve as an estimate of load.

3) Loss-of-function mutations tend to be enriched in sequencing/alignment errors (e.g. MacArthur et al. Science 2012). Thus, other authors have performed systematic checks to safeguard against such errors (e.g. Kleinman-Ruiz PNAS 2022). It would be important to perform similar tests in this study.

Second the statistical reasons:

4) Some of the statistical analyses seem to ignore or not model appropriately a number of well-known complications of analysing fitness variation. Firstly, fitness traits are rarely ever normally distributed (e.g. Bonnet et al. Journal of Heredity 2019, and references therein). Thus, fitness data cannot be modeled using analyses methods that assume a Gaussian distribution, yet this is exactly what the authors are doing. As far as I can tell, all models assume that the dependent variables or the residuals are normally distributed. This assumption seems untenable in this study (see e.g. lines 128-130, which suggest a highly skewed distribution of lifetime reproductive success). The fitness data need to be analysed in a way that accounts for these skewed distributions.

5) The data set for the analyses of mutational load is very small: Whole-genome data and data on reproductive success was available for only 16 foxes, survival was only determined in 14 of the sequenced foxes. This small sample size leads to very low statistical power. Studies of endangered species are - nearly by definition - based on small sample sizes, so this is to be expected. But the low statistical power needs to be addressed explicitly and acknowledged throughout the manuscript. Specifically, I see a number of things that need doing:

A) The statistical power of the analyses needs to be quantified by giving the 95% confidence intervals of the estimates of the effect sizes (e.g. on lines 134-139, 164-170, 203-235). 95% confidence intervals of the effects sizes are a measure of the statistical power of the analyses (Hoenig & Heisey, American Statistician 2001).

B) The differences between the contribution of different types of mutations to fitness variation cannot be assessed by comparing the degrees of statistical significance of separate estimates. For example, comparing the degree of statistical significance of the correlations between loss-of-function mutations and litter size ($r = -0.56$, $p = 0.02$) and between missense mutations and litter size ($r = -0.41$, $p = 0.12$), as currently done in the manuscript (e.g. lines 248-251), is not an appropriate test of the question whether strongly deleterious mutations contribute more to fitness variation than milder mutations. To answer that question, the statistical test needs to ask whether -0.56 is significantly different from -

0.41, given the variation in the estimates of the correlation coefficients and the sample sizes. Meaningful comparisons such as these are currently missing.

C) The statistical power as witnessed by the confidence intervals needs to be considered when interpreting the results. If confidence intervals of the estimated effect sizes are large, then the authors should be much more careful in their interpretations. If confidence intervals of the differences in correlation coefficients overlap zero (as is likely the case with the $r=-0.56$ and $r=-0.41$ comparison above given Figure 1a and d), the appropriate conclusion is not that strongly deleterious mutations are causing most of the observed inbreeding depression, but that both strongly and mildly deleterious mutations seem to contribute to fitness variation, with the data set too small to distinguish the relative magnitudes of the two. Also, with the very small sample size in this study it is imperative that non-significant results are not interpreted as an absence of a relationship (as currently done, e.g. on lines 233 or 248-249). A correlation of -0.41 may not be statistically significant with a sample size of 16, but it may very well be biologically highly significant. Note that I am using the comparison of litter sizes as an example here. The issue applies equally to all other life history traits and all comparisons, not just those of mutational load.

6) Various studies have shown that survival and reproductive success of Arctic foxes are connected to the lemming cycle. Thus, environmental conditions are a major driver of temporal variation in life history traits in these foxes. This complicates the identification of mutations, as the authors note on lines 282-284. Yet, curiously, in the analyses of mutational load this environmental variation is not accounted for. This clearly needs to be done to avoid detecting spurious relationships between mutational load and fitness components. In the analyses of fitness variation as a function of individual ancestry, the phase of the rodent cycle is included as a fixed effect, but the model is not specified explicitly in the manuscript or the Supplementary Material, so the reader doesn't really know what model was fitted. For example, how was the phase of the cycle parameterized? Please explicitly specify the model you fitted.

Detailed comments:

- Figure 2: Panel c) is not mentioned in the caption.
- lines 224-225: This sentence seems to be missing some words.
- line 245: Did I overlook the analysis where it was shown that the effect was not prevalent in heterozygotes? I could not find these analyses.
- line 277: 'combined' would seem more appropriate than 'additive' given that no formal analysis was conducted.
- line 417: Please explain what 'in proximity' means.
- Table S1: Please indicate the exact ancestry of hybrids (F1-F3, etc.) and not just 'hybrid'. Also, please add the year of birth to the table.

Reviewer #3 (Remarks to the Author):

The authors provide a much-needed test of the fitness consequences of deleterious mutations. It has become common to use genomic methods to categorize variants by their likely functional effects. Testing the fitness consequences of these predicted mutational effects is a crucial next step and to my knowledge (and as stated in the manuscript, L253), this manuscript is the first to do so. The authors

also test whether translocated animals for genetic rescue bring genetic load with them and are able to test the fitness consequences of that load in the context of their long-term Arctic fox genetic rescue study.

This manuscript has a lot of potential, however, as written, I find it rather disjointed. Both of the primary questions can advance understanding of genetic rescue, and conservation genomics of small populations in general. However, the authors, in my opinion, end up addressing neither of these objectives completely enough to make them convincing. I think the manuscript would be more convincing if they did a better job explaining the Arctic fox genetic rescue system and addressing the 2nd question thoroughly, then along the way also providing a test of the fitness consequences of the putative mutational effect size assignments. This would de-emphasize the latter results because of the way the data turned out. The paper might not then warrant publication in Nature Communications.

The presentation of empirical data on genetic load brought with translocated foxes into this well-studied populations is really interesting. However, the authors seem forced by space or other constraints to leave important components unexplained. For example, In the methods, it is revealed that of the three translocated foxes, 2 were brothers. From Hasselgren et al. 2018, it appears that both of these brothers had higher reproductive contribution than the third unrelated male. That part of the story seems really important for the overall understanding of how genetic load and inbreeding depression can re-occur. As presented, the reader is left with questions about how the new data fit into our current understanding of this study system.

The first objective, examining the fitness consequences of putatively deleterious variation, was not entirely convincing. Even though the authors resequenced 30 genomes, which is quite a few, the nature of their results does not provide convincing support for the primary conclusions. The primary result is that individuals with a higher proportion of loss of function mutations 'generally' produced smaller litters, lived shorter lives, and had lower lifetime reproductive success. Further, they report no association between moderately deleterious variation and fitness. A founder fox with by far the lowest x-axis values in Fig 1 (a and d) was only included in the analysis of litter size (and not age or LRS). This founder is such a clear outlier and leverage point, the statistical test should be performed with and without her. Even with the author's argument that she had high reproductive contribution and her heterozygous LoF variants were likely expressed as homozygotes in her offspring, their conclusions seem to be completely driven by one individual, at least for average litter size, and the tests should be reported with her excluded. Are the authors comfortable placing so much emphasis with one very high leverage point? If that founder were excluded, or if she happened to have a higher observed value for average litter size by chance (just due to demographic stochasticity), that is, if her observed average litter size happened to be a bit lower, it seems highly likely that the reported relationship would evaporate.

The analysis of age and LRS do not contain that founder because she was excluded for other, valid, reasons. However, there remain high leverage points in these analyses too and unfortunately, the story seems to rely on one individual with low x-axis values and relatively high y-axis values, along with extremely low variation along the x-axis. (how many homozygous LoF genotypes does a difference of 0.001 translate to?)

As a related point, the authors also place too much emphasis on P-values, in my opinion. An r of -0.56 with $p < 0.02$ is deemed to be a 'strong predicted effect on fitness' (for average litter size vs. proportion of homozygous LoF genotypes as an example) and an r of -0.41 ($p = 0.12$) is deemed to be 'no association'. I'm having a hard time with these emphases, especially given the high leverage points.

Thus, while the first test of its kind, my recommendation is to place less emphasis on this portion of the manuscript. It seems too easily dismissed in its current form. However, if included, but with more circumspection and presented as an intriguing first test, I think it becomes more palatable. This would

require a reordering and major reworking of the text, but would actually match the order in which the ideas are currently presented in the introduction.

Related minor points:

L143: as already mentioned, the founder individual had 25% lower mutational load, for proportion of homozygous LoF genotypes, what is the raw difference in number of homozygous genotypes? It would be nice to see this in parentheses, given the proportional difference is roughly 0.0045 vs. 0.0065-ish, or even 0.006 to 0.007ish, just for context.

L62: This discussion would benefit from distinguishing inbreeding from drift load. Larger populations should have a higher number of LEs than small populations, but also have much lower drift load.

L68: it's probably worth noting that the Isle Royale wolf example is also probably not representative of typical wild populations (as was done for the Soay sheep example), since it was one migrant with very high reproductive dominance (how much should we make of this unusual example as a general case for genetic rescue?).

L179: really interesting to see empirical data for LoF as a function of fecundity, but again, I think a rearrangement would be helpful.

L255: The assumptions used by Kyriazis et al. 2021 were strongly questioned by Kardos et al. 2021 (PNAS). It seems caution is warranted in relying on Kyriazis et al. 2021 to make general statements about fitness effects of strongly deleterious alleles, because of their assumed distribution of effect sizes (and mutation rate).

L257: consider replacing 'exerted' with 'exposed'

L266: this paragraph could be shortened due to its speculative nature. Demographic simulations would be really helpful, but perhaps beyond the scope of this ms. One could argue that taking this next step is warranted for this journal, however.

L270: are demographic data available to support the assertion that litter size is expected to be strongly linked to lambda?

L287: The Kardos paper is conspicuously absent from this paragraph. I think those authors made a valuable contribution to the debate that should be included here.

L304: what is 'population development'

L308: again, I suggest caution with statements like this about Kyriazis et al. 2021.

L318: this is really interesting. The idea is that the admixed individuals from 5 small but geographically disparate populations have similar inbreeding load as a large outbred population. This does set up interesting questions in need of testing.

REVIEWER COMMENTS AND AUTHOR REPLIES

Reviewer #1 (Remarks to the Author):

The objectives of this interesting study were to evaluate the relationship between fitness and homozygosity for putatively deleterious alleles, and to determine how ancestry affected the genetic load. The topic is very interesting and important for understanding the genetic basis of inbreeding depression and of how we can use genomic data to understand fitness in small, wild populations of conservation concern. The field data are clearly hard-won and of very high quality and the sequencing and bioinformatic analyses are well done.

There are several issues with the paper, some of which undermine the conclusions:

The finding of reinstatement of inbreeding depression upon backcrossing in either direction is quite interesting. While fitness varies among backcross categories (Figure 2), my general interpretation of the demographic data (at least in part shared by the authors in lines 300-301) is that both natives and immigrants seem to carry similarly substantial loads of deleterious recessive alleles. However, the paper's main focus related to genetic load brought by immigrants is on the genomic data, instead of on the demographic data which actually provide a direct measure of these effects. In particular, there's a big focus on the data shown in Figure S1a (e.g., 2nd to last sentence in the abstract and elsewhere in the results and discussion), which shows that immigrants have a very slightly higher value in some metric of genetic load associated with LOF alleles. The first issue with this is that the effect size, while statistically significant, seems miniscule in magnitude and thus unlikely to be biologically important. The second issue is that I cannot tell how this analysis was actually done. I provide more details on this issue below.

45: You could cite Darwin here, who was keenly interested in inbreeding depression in plants and in red deer estates.

Done (line 44)

47-49, 68-76: The study is set up by suggesting that it is not well understood whether large-effect loci often contribute to inbreeding depression. While the specific distribution of effects is not well understood, it is indeed clear that large effect, deleterious recessive alleles often play a crucial role. I suggest revising the introduction accordingly. There are very good data on this question in pedigree-based analyses of inbreeding depression in domesticated species, and in mice. Those analyses find that inbreeding depression is often largely due to identity-by-descent arising from a small subset of pedigree founders.

If inbreeding depression was purely polygenic (i.e., lots of small-effect loci), then we would expect the contribution of different founders to inbreeding depression to be approximately even. The results from those studies suggest that loci with very large effects (probably a small number of lethal ($s=1$) or semi-lethal alleles (s 'near' 1)) comprise an important fraction of inbreeding depression. We reviewed some of these in our paper that is cited in this section (Kardos et al). I suggest considering those results and citing some of them. A few good citations are listed below, but there are other livestock and *Drosophila* studies that are highly relevant to this section, and also to the statement in lines 74-76.

Lacy et al. (1996, *Evolution* 50: 2187-2200)

Casellas et al. (2009, *J Anim Sciences* 87: 72-79)

Todd et al. (2018, *Scientific Reports* 8: 6167)

With respect to the Isle Royale example focused on strongly here, it is important to consider what is defined as a locus with a large fitness effect ("strongly deleterious" in their paper) compared to large-effect loci that contribute to inbreeding depression in real populations. Robinson et al. defined a strongly deleterious locus as one with $Nes > 100$ (N_e = effective population size and s = the selection coefficient), where N_e was assumed to be 45,000 (left column of their page 7). This means that a 'strongly deleterious' allele had a selection coefficient of at least $s = 100/45,000 = 0.0022$ (~0.2% increase in mortality for a derived allele homozygote compared to a wild type homozygote). I don't think this is a very useful definition of a large-effect locus for the purposes of this study for two reasons. First, alleles with such small effects (s 0.002 or smaller) likely contribute little to inbreeding depression because they are more likely to have nearly additive effects (see citations in Kardos et al. 2021 PNAS). Second, lethal ($s = 1$) and semi-lethal alleles (s of say ~0.5 or larger), i.e. loci with truly very large fitness effects, are commonly observed in real populations and contribute substantially to inbreeding depression where they occur (i.e., highly deleterious alleles are thought to almost always be nearly completely recessive). Two direct observations of lethal recessives in birds are in Trask et al. (2016, *Journal of Animal Ecology*, 85, 879–891) and Ralls et al. (2000, *Animal Conservation*, 3, 145–153.). Consider focusing on empirical examples of the contribution of truly large-effect alleles from the literature that are based on direct demographic inferences (e.g., pedigree-based and direct analyses cited above).

The introduction was mainly focused on genomic studies on wild populations. But the reviewer is right that pedigree studies of captive ones are also highly relevant. We have tried to describe this better, and included the mentioned studies.

54-66: An unmentioned important aspect of this issue is that when there is inbreeding depression, inbred immigrants have to be less likely to survive and reproduce in a new environment than immigrants with substantially higher heterozygosity. There are other

benefits to using immigrants with high heterozygosity, including maximizing adaptive potential (see Ralls et al. 2020).

Agreed. We have now also mentioned positive aspects of immigration from large source populations (line 61-63)

Section beginning at 126. There are several issues that I suggest carefully considering here:

The analysis presented here is meant to test whether homozygosity for LOF alleles (expected to have big fitness effects) is a better predictor of fitness than homozygosity for either missense and nonsynonymous alleles (expected to have small fitness effects, on average). The authors find that fitness is statistically significantly correlated with homozygosity for LOF alleles but not with homozygosity for missense or synonymous alleles, and thus conclude that inbreeding depression is largely driven by large-effect (LOF) alleles (the central finding of the paper: lines 20-25). I think the evidence is presently weaker than suggested in the paper.

There are some important and somewhat complex statistical problems with this analysis. First, the analysis does not account for other variables that are apparently also important (455-458) including temporal or spatial environmental variation and sex, which appear to be accounted for below in the separate analysis of the effects of ancestry on fitness (469-473). Why are these variables accounted for in the analysis of ancestry but no in the analysis testing for effects of homozygosity? Are the environment and sex constant across individuals used in the correlation analysis? That seems unlikely to be so because years of birth vary substantially across the sampled individuals (2001-2018, line 382), but nothing is mentioned in the paper about the distribution of sex among the individuals in the analysis. Assuming sex and environment (e.g., year of birth, natal birth place) vary among individuals in the analysis, then it seems they should be accounted for statistically. The sample size may be too small for the analysis to be very informative.

Second, it would be unsurprising if the difference in the observed correlations between fitness components versus homozygosity measured with different subsets of loci (LOF versus missense versus nonsynonymous loci) is due solely to a necessarily high sampling variance in the estimation of the correlation coefficients. With respect just to statistical phenomena, both p-values and regression coefficients are highly stochastic when sample sizes are so small. When inbreeding depression is present, we expect fitness (w) to be negatively correlated (on average) with homozygosity (H) measured at any set of polymorphic loci in the genome. This is expected even when we do not directly measure homozygosity at the loci that cause inbreeding depression, because of genome-wide identity disequilibrium that occurs when inbreeding varies among analyzed individuals. See several relevant papers from Patrice David, especially Szulkin et al. (2010, *Evolution*) which explains in detail an analogous statistical problem of testing for local effects

driving heterozygosity-fitness correlations. The correlation coefficient (r) from a regression of w on H will vary across different sets of loci just due to sampling variation. Holding the strength of inbreeding depression constant, the sampling variance in r is expected to be higher when fewer loci are genotyped, minor allele frequency is lower (as is the case with LOF loci), and when fewer individuals are analyzed (sample sizes here are extremely small). All three of these components of the sampling variance (sample size, number of loci, minor allele frequency) are important here, but do not appear to be accounted for statistically when evaluating the observed differences among models. The relevant question is not whether r varies among models that use different subsets of loci to measure H (they always do), or whether one is statistically significant and the other isn't, but rather whether the regressions vary more than expected by chance. I suggest modifying the statistical analysis to test whether the stronger correlation observed when H is measured with LOF variants compared to missense or nonsynonymous variants (the key finding of the paper, assuming this holds when other important variables are accounted for) is likely to be due to simple sampling variation in r among different sets of loci analyzed. I believe an F-ratio test could be used for this (see a similar application in Szulkin et al. (2010) from the pre-genomics era), but it seems that would only account for sampling variance arising from small sample size and would ignore effects of number of loci and allele frequency distributions. It might be possible to devise a permutation test, where the null distribution of the difference in r between models is based on permuted loci, which would also account for sampling variance arising from number of loci and allele frequency differences. Again the extremely small sample sizes are a major limitation.

Lastly, restricting this analysis to include only coding loci with LOF, missense, and synonymous mutations means that one of the most important comparisons is not made. The coding part of genomes represents a small fraction of the loci that are subjected to purifying selection (e.g., <https://doi.org/10.1371/journal.pgen.1004525>). I therefore suggest comparing the association of fitness with homozygosity for LOF/missense/synonymous alleles versus homozygosity across the whole genome (i.e., homozygosity at all nearly 11,000,000 SNPs, line 414) which accounts for deleterious recessive effects across the entire genome.

Thank you for this valuable comment. We agree that the sample size is small. Therefore we do not have the same statistical power as in the larger dataset on demography to control for all environmental factors. However, we have now managed to extract fitness data for 7 additional individuals from a more northwards located subpopulation in Sweden, with similar population history. This has increased the power so we can control for which phase of the rodent cycle each individual was born during. We find that individuals with a large proportion of homozygous LOFs live shorter lives and have lower lifetime reproductive success, while the average litter size does no longer significantly differ when controlling for rodent phase. We find a similar result when also controlling

for sex of each individual but we don't believe it is reasonable to add a third variable based on this small dataset. We have also toned down on statement that moderately deleterious mutations don't drive inbreeding depression, as our small dataset can't really rule this out.

Regarding homozygosity throughout the genome, we have explored the link with fitness (juvenile survival) in a previously published paper (Hasselgren et al. 2021), but if you still wish we would be happy to include this in this manuscript too.

With our data we find that FROH is highly correlated with both the proportion of missense alleles and synonymous alleles in homozygous state, however it is not well correlated with the homozygous LoFs, implying that selection may exert a strong effect on them. This data could be included in the manuscript, but is not currently included.

163-167: Are the relevant methods for this in lines 428-430? It's unclear exactly what was done so I cannot interpret this result. What I think is the relevant method description is extremely short and says:

"We calculated the proportion of variants belonging to each category by dividing the number of alleles in respective category with the total number of alleles (LoF + moderate + synonymous). We also took genotypes in heterozygous and homozygous state into account."

This first refers to 'proportion of variants' which is the same thing as proportion of loci, but then says this is done by counting alleles, which is not the same as counting loci. I suggest defining this explicitly with an equation and clearly defined terms. Additionally, I cannot tell what is meant by "We took genotypes in heterozygous and homozygous state into account". It might be clearer to address topic in the paper using the standard R_x/y method of Do et al. (Nature Genetics 47, 126-131 (2015)), which would work well if the efficiency of historical purifying selection in one population versus the other is the focus of interest here. I cannot tell how it was calculated that "immigrant descendants had 4 % higher proportion of LoF alleles compared with natives" given the very small difference that is clear in figure S1A: ~ 0.0088 (native median) compared to ~ 0.0090 (immigrant descendants). It is hard to see this very small difference as being biologically important, but it is a major focus of the paper (see 2nd to last sentence in the abstract), but then again, I don't really understand how the analysis was done so it's impossible for me to interpret it properly.

The concepts of variants and alleles were a bit mixed up in the previous version.

We have now tried to clarify how we calculated the proportion of alleles or genotypes in each category (line 465-474). Similar approaches have been carried out in other studies. E.g. Vallaru et al. 2019.

Regarding the 4% difference in deleterious variation, it was calculated as the difference in the proportion of LoF alleles in natives (0,008665) and hybrids (0,009008). We have now replaced this to describe the total number of LoF variants in the native (1168 variants) versus hybrid (1220 variants) gene pool. Although the difference may seem small, even a small number of LoFs introduced could potentially have a pronounced effect on fitness.

240-242: There are many more examples from genetic rescue studies.

These are examples of studies that have linked genome wide inbreeding to fitness, we do not aim to cover all of the studies here (hence the "e.g."). Two more studies have now been added (line 275).

250: An insurmountable problem for analyses like this is that we do not know the distribution of the effects (selection coefficient) for any mutation class.

Yes that is true, this sentence has been rephrased. We are no longer comparing the effect between missense and LoF alleles (line 282-285).

252-254: The statistical evidence for this finding is insufficient in my opinion, as described above in detail.

The statistics have been re-run, adding additional samples and controlling for rodent abundance (see methods).

256: And consistent with lots of previous empirical data as noted above.

The sentence this comment is referring to has been removed.

289: I suggest defining 'common' here. "16 LoF alleles that were common in immigrant descendants but absent in natives" represents about 1.5% of the total of 1,052 detected LOF alleles. It's a bit difficult to interpret this is being highly biologically important.

We do not mean that "16 out of 1052" is common. We mean that these 16 LoFs that were absent in natives had relatively high *frequency*. The frequency of these 16 LoFs were 20-37%, this had been added to the results section (line 206).

301-303: Is this based on the observation of "16 LoF alleles that were common in immigrant descendants but absent in natives"?

We found 132 LoFs in hybrids that were not present in natives, of these, 16 were relatively common. Moreover, even a small number of deleterious mutations could potentially have a large effect on fitness.

455-458: Please clarify whether the predictor for each individual is calculated as the proportion of genotyped LOF/missense/synonymous loci where the individual is homozygous for the DERIVED allele. This is an important detail that appears to be missing from them manuscript. I also suggest that it would be useful to include a figure somewhere comparing the derived allele frequencies of derived LOF/missense/synonymous/ alleles (i.e., the sfs), which would help readers to understand why homozygosity is so much lower for LOF alleles than for missense alleles (x-axes in upper panels of Figure 1). If this has been done in a previous paper, it would be sufficient to state that the SFS are different and refer to the previous study.

Yes it is the derived allele compared to the red fox assembly. This information has been added (line 465). An SFS has also been added to the supplementary material.

I hope the authors find my comments helpful!
Marty Kardos

Reviewer #2 (Remarks to the Author):

Strongly deleterious mutations influence reproductive output and longevity in an endangered population

This manuscript addresses an important question in conservation biology: what role do deleterious mutations play in generating fitness variation, which - in turn - affects the long-term survival of endangered species? This question is addressed using empirical data from a small population of the endangered Arctic fox in Sweden, which, despite its geographical isolation does experience occasional immigration.

The main findings are that foxes with a high proportion of loss-of-function mutations live less long and produce fewer offspring, and that offspring of immigrants had on average 4% more of these loss-of-function variants. The latter finding suggests that immigrants brought in deleterious mutations into this endangered fox population.

While this work asks very interesting and important questions, it is difficult to know what

to make of the results for a number of reasons. I will divide these reasons into biological and into statistical reasons.

First the biological reasons:

1) The major difficulty with this data set is the recent demographic history of this Arctic fox population. From 2001 to 2011 the population size increase from ~5 breeding adults to ~50, then crashing to ~5 breeders in 2016 before returning to ~30-40 in 2017-2019. In parallel, the average inbreeding in the population increased by a factor of ~10. Following the immigration of 3 males in 2010-2011, average inbreeding declined, although not as much as one might have expected because there was extensive inbreeding within the immigrant lineages, including parent-offspring matings (all based on Lotsander et al. 2021). Thus, effective population size, average inbreeding, proportion of immigrant ancestry, phase of the lemming cycle, and probably several other factors related to fitness (such as those reported in previous publications by the authors, e.g. Erlandsson et al. 2017 or Choi et al. 2019) all covary systematically with each other. This makes it extremely challenging to quantify the contribution of deleterious mutations to fitness variation, because a given deleterious mutation is likely present in homozygous state only during one particular phase of the lemming cycle in one or a few individuals with certain ancestries. Thus, separating environmental effects from genetic effects seems nigh impossible, especially with a sample size of 16 and 14, respectively. At the very least, the confounding of inbreeding, immigrant ancestry, and environmental factors would need to be explored in much more detail in the manuscript or the supplementary material before one can have confidence in the results.

We agree, it is a complex study system where environmental effects play an important role to individual fitness. We have now managed to add seven additional individuals to the analysis on mutational load and fitness and could therefore also control for rodent abundance each year (line 504-507. See comment from reviewer one.

2) Unfortunately, one of the main genomic measures used, the proportion of homozygote genotypes of a particular class, is never defined explicitly. I assume it equates to the number of homozygotes of a particular class (e.g. loss-of-function) divided by the total number of variable sites of this class. If this is correct, then the numbers on the x axes in Figure 1 suggest that for LoF variants the range of number of homozygous mutations between the two most extreme individuals is $0.005 \times 1052 = 5.26$ and $0.007 \times 1052 = 7.36$, and for missense variants 9330 and 10664, respectively. Would this not argue for a far more important role of the mildly deleterious missense mutations than the LoF mutations? A similar reasoning would suggest that a regression of fitness traits on the proportion of homozygotes would be the more appropriate measure of mutational load than the unitless correlation. I don't think a unitless measure can serve as an estimate of load.

The proportion of homozygous genotypes were calculated as the number of homozygous LoF genotypes by the total number of scored genotypes, that is both LoFs, missense and synonymous genotypes. We have tried to clarify this (line 465-474).

We now also use regression models rather than correlation tests (line 498).

3) Loss-of-function mutations tend to be enriched in sequencing/alignment errors (e.g. MacArthur et al. Science 2012). Thus, other authors have performed systematic checks to safeguard against such errors (e.g. Kleinman-Ruiz PNAS 2022). It would be important to perform similar tests in this study.

We have checked all LoFs manually by comparing the genes of the red fox assembly with the vcf-file of each individual to confirm that the variant is actually derived relative to the assembly. This information has been added to the manuscript (line 414-423; 461-463).

Second the statistical reasons:

4) Some of the statistical analyses seem to ignore or not model appropriately a number of well-known complications of analysing fitness variation. Firstly, fitness traits are rarely ever normally distributed (e.g. Bonnet et al. Journal of Heredity 2019, and references therein). Thus, fitness data cannot be modeled using analyses methods that assume a Gaussian distribution, yet this is exactly what the authors are doing. As far as I can tell, all models assume that the dependent variables or the residuals are normally distributed. This assumption seems untenable in this study (see e.g. lines 128-130, which suggest a highly skewed distribution of lifetime reproductive success). The fitness data need to be analysed in a way that accounts for these skewed distributions.

Age and LRS were not normally distributed. We performed box cox and normality tests and according to these we have now log transformed longevity and square rooted LRS (line 501-504).

5) The data set for the analyses of mutational load is very small: Whole-genome data and data on reproductive success was available for only 16 foxes, survival was only determined in 14 of the sequenced foxes. This small sample size leads to very low statistical power. Studies of endangered species are - nearly by definition - based on small sample sizes, so this is to be expected. But the low statistical power needs to be addressed explicitly and acknowledged throughout the manuscript. Specifically, I see a number of things that need doing:

A) The statistical power of the analyses needs to be quantified by giving the 95% confidence intervals of the estimates of the effect sizes (e.g. on lines, 203-235). 95%

confidence intervals of the effects sizes are a measure of the statistical power of the analyses (Hoenig & Heisey, American Statistician 2001).

We have now added confidence intervals for all statistical tests performed, see results section.

B) The differences between the contribution of different types of mutations to fitness variation cannot be assessed by comparing the degrees of statistical significance of separate estimates. For example, comparing the degree of statistical significance of the correlations between loss-of-function mutations and litter size ($r=-0.56$, $p=0.02$) and between missense mutations and litter size ($r=-0.41$, $p=0.12$), as currently done in the manuscript (e.g. lines 248-251), is not an appropriate test of the question whether strongly deleterious mutations contribute more to fitness variation than milder mutations. To answer that question, the statistical test needs to ask whether -0.56 is significantly different from -0.41 , given the variation in the estimates of the correlation coefficients and the sample sizes. Meaningful comparisons such as these are currently missing.

We are no longer stating that LoF mutations contribute more than missense mutations to inbreeding depression as we have not explicitly tested this.

C) The statistical power as witnessed by the confidence intervals needs to be considered when interpreting the results. If confidence intervals of the estimated effect sizes are large, then the authors should be much more careful in their interpretations. If confidence intervals of the differences in correlation coefficients overlap zero (as is likely the case with the $r=-0.56$ and $r=-0.41$ comparison above given Figure 1a and d), the appropriate conclusion is not that strongly deleterious mutations are causing most of the observed inbreeding depression, but that both strongly and mildly deleterious mutations seem to contribute to fitness variation, with the data set too small to distinguish the relative magnitudes of the two. Also, with the very small sample size in this study it is imperative that non-significant results are not interpreted as an absence of a relationship (as currently done, e.g. on lines 233 or 248-249). A correlation of -0.41 may not be statistically significant with a sample size of 16, but it may very well be biologically highly significant. Note that I am using the comparison of litter sizes as an example here. The issue applies equally to all other life history traits and all comparisons, not just those of mutational load.

We agree. We have added confidence intervals to all statistical tests and are no longer stating that missense mutations do not contribute to inbreeding depression.

6) Various studies have shown that survival and reproductive success of Arctic foxes are connected to the lemming cycle. Thus, environmental conditions are a major driver of temporal variation in life history traits in these foxes. This complicates the identification

of mutations, as the authors note on lines 282-284. Yet, curiously, in the analyses of mutational load this environmental variation is not accounted for. This clearly needs to be done to avoid detecting spurious relationships between mutational load and fitness components. In the analyses of fitness variation as a function of individual ancestry, the phase of the rodent cycle is included as a fixed effect, but the model is not specified explicitly in the manuscript or the Supplementary Material, so the reader doesn't really know what model was fitted. For example, how was the phase of the cycle parameterized? Please explicitly specify the model you fitted.

In the revised version of the manuscript we have now controlled for the phase of the rodent cycle in the analysis of mutational load and fitness. In the previous version, the sample size was too small to do so but we have now retrieved fitness measures for seven additional foxes (see replies in previous comments).

The models fitted and their outputs have been added to the supplementary material. We have also explained better how the phases of the rodent cycle were used in the analyses (line 504-507).

Detailed comments:

- Figure 2: Panel c) is not mentioned in the caption.

Fixed

- lines 224-225: This sentence seems to be missing some words.

Yes, we have now changed this sentence slightly.

- line 245: Did I overlook the analysis where it was shown that the effect was not prevalent in heterozygotes? I could not find these analyses.

This part must have been accidentally deleted in some way in the previous version, we apologize for this. We have now added the results on the analysis of the effect of heterozygote LoFs on fitness (line 147-150).

- line 277: 'combined' would seem more appropriate than 'additive' given that no formal analysis was conducted.

Changed accordingly

- line 417: Please explain what 'in proximity' means.

"In proximity" was added by a mistake, we have only looked at mutations within coding genes. Modifier mutations can be found in proximity to coding genes (5Kb) from a gene but we did not include this category since it is very hard to predict whether they will have

an impact in reality. We have now removed "in proximity" (line 451).

- Table S1: Please indicate the exact ancestry of hybrids (F1-F3, etc.) and not just 'hybrid'. Also, please add the year of birth to the table.

Added

Reviewer #3 (Remarks to the Author):

The authors provide a much-needed test of the fitness consequences of deleterious mutations. It has become common to use genomic methods to categorize variants by their likely functional effects. Testing the fitness consequences of these predicted mutational effects is a crucial next step and to my knowledge (and as stated in the manuscript, L253), this manuscript is the first to do so. The authors also test whether translocated animals for genetic rescue bring genetic load with them and are able to test the fitness consequences of that load in the context of their long-term Arctic fox genetic rescue study.

This manuscript has a lot of potential, however, as written, I find it rather disjointed. Both of the primary questions can advance understanding of genetic rescue, and conservation genomics of small populations in general. However, the authors, in my opinion, end up addressing neither of these objectives completely enough to make them convincing. I think the manuscript would be more convincing if they did a better job explaining the Arctic fox genetic rescue system and addressing the 2nd question thoroughly, then along the way also providing a test of the fitness consequences of the putative mutational effect size assignments. This would de-emphasize the latter results because of the way the data turned out. The paper might not then warrant publication in Nature Communications.

The presentation of empirical data on genetic load brought with translocated foxes into this well-studied populations is really interesting. However, the authors seem forced by space or other constraints to leave important components unexplained. For example, In the methods, it is revealed that of the three translocated foxes, 2 were brothers. From Hasselgren et al. 2018, it appears that both of these brothers had higher reproductive contribution than the third unrelated male. That part of the story seems really important for the overall understanding of how genetic load and inbreeding depression can re-occur. As presented, the reader is left with questions about how the new data fit into our current understanding of this study system.

It was revealed in Lotsander et al. 2021 that the unrelated immigrant was in fact also rather successful, although the brothers were indeed more successful. We have added their respective proportion of ancestry in the population to the manuscript (line 378-380).

The first objective, examining the fitness consequences of putatively deleterious variation,

was not entirely convincing. Even though the authors resequenced 30 genomes, which is quite a few, the nature of their results does not provide convincing support for the primary conclusions. The primary result is that individuals with a higher proportion of loss of function mutations 'generally' produced smaller litters, lived shorter lives, and had lower lifetime reproductive success. Further, they report no association between moderately deleterious variation and fitness. A founder fox with by far the lowest x-axis values in Fig 1 (a and d) was only included in the analysis of litter size (and not age or LRS). This founder is such a clear outlier and leverage point, the statistical test should be performed with and without her. Even with the author's argument that she had high reproductive contribution and her heterozygous LoF variants were likely expressed as homozygotes in her offspring, their conclusions seem to be completely driven by one individual, at least for average litter size, and the tests should be reported with her excluded. Are the authors comfortable placing so much emphasis with one very high leverage point? If that founder were excluded, or if she happened to have a higher observed value for average litter size by chance (just due to demographic stochasticity), that is, if her observed average litter size happened to be a bit lower, it seems highly likely that the reported relationship would evaporate.

The analysis of age and LRS do not contain that founder because she was excluded for other, valid, reasons. However, there remain high leverage points in these analyses too and unfortunately, the story seems to rely on one individual with low x-axis values and relatively high y-axis values, along with extremely low variation along the x-axis. (how many homozygous LoF genotypes does a difference of 0.001 translate to?)

Thank you for these valuable comments, which are also similar to the ones that reviewer 1 & 2 also pointed out. To account for this, to gain more power in this analysis, we have included seven additional foxes. To account for demographic stochasticity we have controlled for whether an individual was born during increasing or decreasing rodent abundance. We have also transformed the fitness characters LRS and age (see methods section).

As a related point, the authors also place too much emphasis on P-values, in my opinion. An r of -0.56 with $p < 0.02$ is deemed to be a 'strong predicted effect on fitness' (for average litter size vs. proportion of homozygous LoF genotypes as an example) and an r of -0.41 ($p = 0.12$) is deemed to be 'no association'. I'm having a hard time with these emphases, especially given the high leverage points.

Thus, while the first test of its kind, my recommendation is to place less emphasis on this portion of the manuscript. It seems too easily dismissed in its current form. However, if included, but with more circumspection and presented as an intriguing first test, I think it becomes more palatable

. This would require a reordering and major reworking of the text, but would actually match the order in which the ideas are currently presented in the introduction.

This is also a valid point. We have toned down on the statement that expression of LoFs drive inbreeding depression rather than moderately deleterious mutations. However, we only state that LoFs in general are predicted to have a strong effect on fitness, which has strong support in theory. Moreover, we have changed "no effect" to "no significant effect" on fitness.

Related minor points:

L143: as already mentioned, the founder individual had 25% lower mutational load, for proportion of homozygous LoF genotypes, what is the raw difference in number of homozygous genotypes? It would be nice to see this in parentheses, given the proportional difference is roughly 0.0045 vs. 0.0065-ish, or even 0.006 to 0.007ish, just for context.

We have now added this information (line 166-168).

L62: This discussion would benefit from distinguishing inbreeding from drift load. Larger populations should have a higher number of LEs than small populations, but also have much lower drift load.

Yes, this is a good point and has been added to the introduction (line 61-63)

L68: it's probably worth noting that the Isle Royale wolf example is also probably not representative of typical wild populations (as was done for the Soay sheep example), since it was one migrant with very high reproductive dominance (how much should we make of this unusual example as a general case for genetic rescue?).

This is a valid point. We have now removed the sentence about Soay sheep not being a representative example, rather than adding an additional one about the wolf population, in order to make the introduction more stringent.

L179: really interesting to see empirical data for LoF as a function of fecundity, but again, I think a rearrangement would be helpful.

As mentioned above, we have added some additional individuals to the fitness analyses to gain more statistical power (line 413-422).

L255: The assumptions used by Kyriazis et al. 2021 were strongly questioned by Kardos et al. 2021 (PNAS). It seems caution is warranted in relying on Kyriazis et al. 2021 to make general statements about fitness effects of strongly deleterious alleles, because of their assumed distribution of effect sizes (and mutation rate).

This sentence has been removed.

L257: consider replacing 'exerted' with 'exposed'

Changed accordingly (line 287)

L266: this paragraph could be shortened due to its speculative nature. Demographic simulations would be really helpful, but perhaps beyond the scope of this ms. One could argue that taking this next step is warranted for this journal, however.

This paragraph has been shortened a bit (line 298-314)

L270: are demographic data available to support the assertion that litter size is expected to be strongly linked to lambda?

This is a valid point, there are no studies explicitly studying the effect of litter size on lambda in the Scandinavian arctic fox. This study is however using litter size as a factor when quantifying lambda:

<https://doi.org/10.1111/j.1365-2664.2008.01515.x>

However, when we shortened this paragraph we removed the statement that smaller litter sizes will reduce "population long term viability".

L287: The Kardos paper is conspicuously absent from this paragraph. I think those authors made a valuable contribution to the debate that should be included here.

We agree and have added this paper to both the introduction and discussion (line 63; 317).

L304: what is 'population development'

We rephrased to "population size decreased" (line 334-335)

L308: again, I suggest caution with statements like this about Kyriazis et al. 2021.

We have changed this reference to two classical papers that suggest the same (line 340)

L318: this is really interesting. The idea is that the admixed individuals from 5 small but geographically disparate populations have similar inbreeding load as a large outbred population. This does set up interesting questions in need of testing.

Agreed, although outside the scope of this manuscript we are aiming to test this theory in the near future.

REVIEWER COMMENTS

Reviewer #1 (Remarks to the Author):

I remain enthusiastic and believe the paper will be widely read and highly cited once it is published. The revision has improved the paper, but several issues remain. In summary, I believe a further revision should include the following:

- clarification of the definition of the different load metrics in the result section. The metrics of load are currently quite unclear, which makes it very difficult to follow what has been done and what the results actually mean. I think it would help tremendously to begin each results section with a clear description of the motivation and definition of the relevant metrics of load.
- The biological relevance of the apparently slightly higher burden of LOF alleles among individuals with immigrant ancestry compared to pure native foxes (4.5 %) remains totally unclear. I still suspect that it is of little biological significance, but I would be very happy to be proven wrong. The size of these effects should either be estimated explicitly, or downplayed in the narrative throughout the paper.
- The explanation for not including sex as a predictor in the analysis of the effects of genetic load metrics on fitness components seems inadequate to me.

Details:

Throughout (e.g., 24-25; 196-199; 332-334): Again (as in the first round of review), the difference in 'mutational load' between natives and immigrant descendants seems extremely tiny (Figure S2). The authors say in their rebuttal that this small difference might mean a large difference in fitness, and this perspective is implied throughout the revised MS, but I still do not know what the basis for this view is. To remedy this, I strongly suggest either estimating the fitness effect of the difference in mutational load given your current statistical results (if they are indeed compatible), recognizing that LOF alleles are likely to have mostly recessive fitness effects (consistent with your results). Alternatively, you could describe in detail why you believe a 4.5% higher burden of LOF alleles in individuals with immigrant ancestry is likely to have a substantive impact on population fitness; this will be hard to do convincingly absent a quantitative analysis. Otherwise, the narrative still seems to me to be making a mountain out of a mole hill.

It remains unclear to me why sex was not included in the analysis of the fitness effects of LOF alleles, but it was included in the analysis of effects of ancestry. The rebuttal says that the authors didn't think it was reasonable to account for a third predictor variable given the small sample size, but there the readers of the published paper will still question this as I and I believe another reviewer did. I suggest either adding sex as a predictor to the models described in lines 152-158, which account for rodent phase, or instead explain in the methods why you excluded it in this case but included it when analyzing effects of ancestry on fitness. I also want to be clear that I think it is fine to show results for analyses where the confounders are and aren't controlled for.

In their rebuttal the authors wrote "With our data we find that FROH is highly correlated with both the proportion of missense alleles and synonymous alleles in homozygous state, however it is not well correlated with the homozygous LoFs, implying that selection may exert a strong effect on them. This data could be included in the manuscript, but is not currently included."

Is there a reason to not show an analysis of the relationship between genome-wide homozygosity and the fitness components? The comparison between the results from such an analysis and those that used specific load metrics in the current paper is an important one as the former integrates effects from across the entire genome, and the latter accounts for effects of loci that we 'think' are likely to be most important. I don't think this is essential, but it would strengthen the paper. The lack of a relationship between Froh and LOF homozygotes may just be because the average number of LOF

homozygous alleles within individuals is small (whereas this number is extremely large with synonymous mutations I suspect). There were 1,300 LOF polymorphisms detected in the population, but what is the average number LOF homozygous alleles/individual? I doubt that selection is the mechanism as it would require a gigantic response to selection over a very short time frame.

24-25: Can the statistical results in the current paper be used to test whether 4.5 % more LOFs would significantly impact fitness? I remain unconvinced that this difference is biologically significant.

78-80: I described in detail in my original review that the 'strongly deleterious alleles' as defined by Robinson et al. had an selection coefficient as small as $s = 0.0022$ ($\sim 0.2\%$ increase in mortality for a derived allele homozygote compared to a wild type homozygote), but there was no response. I strongly suggest either removing this statement altogether, or adding a description of why you agree with Robinson et al. that a locus with $s = 0.002$ qualifies as 'strongly deleterious' against the backdrop of much larger effect (i.e., lethal) deleterious mutations being detected commonly in real populations.

Throughout: The word 'significant' is used throughout in reference to statistical results. I suggest changing this to 'statistically significant' to avoid confusion regarding biological versus statistical significance.

191-193: This is not clearly written. Any reader paying attention will wonder what is the 'proportion of LOF mutations'? Is this the proportion of all LOF alleles where an individual has at least one copy? Or are you still referring to proportion of homozygous LOF alleles as above? Please define each metric explicitly in the results. As currently written, this lack of clarity makes it quite hard to not lose track of what has been done from one section to the next. Perhaps start out each section with a clear description of the relevant metric of load and why it was used.

196: I think this should be figure S2a

Section beginning at 204: It would help to point out here the caveat that we have no idea what are the fitness impact of any of these LOF alleles. There will always be alleles that satisfy the criteria used here to identify candidate LOF alleles, even when there are no fitness effects at all.

316: it is unclear what 'genomic composition' means.

338-342: This is true for segregating deleterious alleles. However, populations that are small for a long time are expected to have more fixed deleterious alleles (and often lower intrinsic fitness in one to one comparisons) than historically larger ones.

Marty Kardos

Reviewer #4 (Remarks to the Author):

The authors addressed many aspects of the comments of reviewer 2. However, in my opinion there are still few adjustments to be made. In the attached file the comments are added in blue to the former comments of reviewer 2.

Comment to comment 1:

I appreciate that 7 individuals could be added to the analyses of mutational load and that the rodent cycle could be considered in the linear regression. However, I would suggest to only include the regression model with the rodent cycle considered in the main text and in Figure 1. The outcome of the regression should be shown in detail in the supplementary file as it is done for the ancestry analyses. Litter size is not anymore significantly influenced by proportion of homozygous LoF

genotypes when considering rodent cycle. This should be accounted for in the discussion (line 301-303).

The knowledge that ancestry influences the fitness parameters as shown in the ancestry analysis (with more individuals) suggests that it should be at least mentioned why you do not include the immigration ancestry in the analyses of mutational load or show the model with ancestry included in the supplementary file. This might imply to first show the analyses of ancestry and then the analyses of mutational load.

The confounding factor of inbreeding was is not yet addressed/discussed. See comments 5B below.

Comment to comment 5B:

I concur with reviewer 1 that a comparison of the effect size of the regression with the proportion of homozygous LoF genotypes with a regression of overall homozygosity from all markers or FROH would be appreciated at least in the supplementary file to get more insight into the difference in the effect of overall inbreeding, homozygous LoFs and homozygous missense genotypes.

Please mention the correlation of FROH with the proportion of LoFs, missense alleles and synonymous alleles in homozygous state that you mentioned in the answer to reviewer 1, also at least in the supplementary file in case you are already at the limit of number of words in the main text.

Minor comments:

Line 143: It should be LoF genotypes in homozygous state

Line 174: correct spelling of mutations

Line 196: It should be fig. S2a

Figure 2c: Shouldn't the legend of y-axis be "litter size" instead of "average litter size"?

Reviewer #1 (Remarks to the Author):

I remain enthusiastic and believe the paper will be widely read and highly cited once it is published. The revision has improved the paper, but several issues remain. In summary, I believe a further revision should include the following:

- clarification of the definition of the different load metrics in the result section. The metrics of load are currently quite unclear, which makes it very difficult to follow what has been done and what the results actually mean. I think it would help tremendously to begin each results section with a clear description of the motivation and definition of the relevant metrics of load.
- The biological relevance of the apparently slightly higher burden of LOF alleles among individuals with immigrant ancestry compared to pure native foxes (4.5 %) remains totally unclear. I still suspect that it is of little biological significance, but I would be very happy to be proven wrong. The size of these effects should either be estimated explicitly, or downplayed in the narrative throughout the paper.
- The explanation for not including sex as a predictor in the analysis of the effects of genetic load metrics on fitness components seems inadequate to me.

Details:

Throughout (e.g., 24-25; 196-199; 332-334): Again (as in the first round of review), the difference in 'mutational load' between natives and immigrant descendants seems extremely tiny (Figure S2). The authors say in their rebuttal that this small difference might mean a large difference in fitness, and this perspective is implied throughout the revised MS, but I still do not know what the basis for this view is. To remedy this, I strongly suggest either estimating the fitness effect of the difference in mutational load given your current statistical results (if they are indeed compatible), recognizing that LOF alleles are likely to have mostly recessive fitness effects (consistent with your results). Alternatively, you could describe in detail why you believe a 4.5% higher burden of LOF alleles in individuals with immigrant ancestry is likely to have a substantive impact on population fitness; this will be hard to do convincingly absent a quantitative analysis. Otherwise, the narrative still seems to me to be making a mountain out of a mole hill.

Thanks for this comment. The idea was that even if the number of introduced LoFs is small, the homozygosity of even a small number of LoFs could impact fitness if they are located in genes important to fitness. However, it is indeed true that we were only analyzing the total proportion of LoF alleles between natives and immigrants and not those in homozygote state. We now performed a non-parametric anova and found that there is no statistically significant difference in proportion of homozygous LoFs between ancestry groups (native, F1 and F2+F3; not included in the manuscript) and hence we have toned down on the statement that the introduced LoFs cause inbreeding

depression. We have removed the sentence about mutational load in immigrants vs natives from the abstract in order to not put too much emphasis on this. We have also removed the statement from the discussion that the higher load in immigrant descendants may lead to inbreeding depression in additional traits than prior to immigration. We also write in the discussion that it is noteworthy that the effect size of 4.5% is small (**line 330-333**).

It remains unclear to me why sex was not included in the analysis of the fitness effects of LOF alleles, but it was included in the analysis of effects of ancestry. The rebuttal says that the authors didn't think it was reasonable to account for a third predictor variable given the small sample size, but there the readers of the published paper will still question this as I and I believe another reviewer did. I suggest either adding sex as a predictor to the models described in lines 152-158, which account for rodent phase, or instead explain in the methods why you excluded it in this case but included it when analyzing effects of ancestry on fitness. I also want to be clear that I think it is fine to show results for analyses where the confounders are and aren't controlled for.

We have now included sex as a parameter in the analyses on genomic variation and fitness. In accordance with the comments from reviewer 2 we have chosen to include only the statistics where we control for rodent phase (and now also sex). Thus, we have removed the results where fitness was an effect of only mutational load. **Line: 141-142, 172-173, 528-530.**

In their rebuttal the authors wrote "With our data we find that FROH is highly correlated with both the proportion of missense alleles and synonymous alleles in homozygous state, however it is not well correlated with the homozygous LoFs, implying that selection may exert a strong effect on them. This data could be included in the manuscript, but is not currently included." Is there a reason to not show an analysis of the relationship between genome-wide homozygosity and the fitness components? The comparison between the results from such an analysis and those that used specific load metrics in the current paper is an important one as the former integrates effects from across the entire genome, and the latter accounts for effects of loci that we 'think' are likely to be most important. I don't think this is essential, but it would strengthen the paper. The lack of a relationship between Froh and LOF homozygotes may just be because the average number of LOF homozygous alleles within individuals is small (whereas this number is extremely large with synonymous mutations I suspect). There were 1,300 LOF polymorphisms detected in the population, but what is the average number LOF homozygous alleles/individual? I doubt that selection is the mechanism as it would require a gigantic response to selection over a very short time frame.

The average number of homozygous LoFs are ~500 per individual. We have now added genome wide heterozygosity (number of heterozygous sites per kb) as a parameter to the manuscript, investigating its relationship with fitness. We found no statistically

significant relationship with LRS, litter size or longevity. However, we also decided to add the fitness trait juvenile survival which we have previously found to be connected to inbreeding and heterozygosity. We find that there is a significant relationship between genome wide heterozygosity and survival. We also find a relationship between missense and synonymous homozygous mutations and survival, and these metrics are highly correlated with genome wide heterozygosity. Juvenile survival is however not significantly correlated with homozygous LoFs. We reason that there are different genomic regions involved in these different fitness traits. **Fig. 1, lines 23-28, 102-105, 157-158, 166-169, 287-295, 300-303, 416-418, 507-516, 523-526.**

24-25: Can the statistical results in the current paper be used to test whether 4.5 % more LOFs would significantly impact fitness? I remain unconvinced that this difference is biologically significant.

As this comparison was made for the total number of LoFs (alleles both in homozygous and heterozygote state) it is difficult to test the fitness effect. Since we did not find a significant difference in the proportion of homozygous LoFs between the groups we decided to tone down on the fitness impacts this might have on the population.

78-80: I described in detail in my original review that the 'strongly deleterious alleles' as defined by Robinson et al. had an selection coefficient as small as $s = 0.0022$ (~0.2% increase in mortality for a derived allele homozygote compared to a wild type homozygote), but there was no response. I strongly suggest either removing this statement altogether, or adding a description of why you agree with Robinson et al. that a locus with $s = 0.002$ qualifies as 'strongly deleterious' against the backdrop of much larger effect (i.e., lethal) deleterious mutations being detected commonly in real populations.

We are sorry for the lack of response to this comment, it was part of a longer comment and we misinterpreted it to advising to focus the introduction also on empirical pedigree based studies rather than only on wild population. Reading the comment again it is evident that this was a misinterpretation. It is true that a selection coefficient of 0.0022 is very small. We have thus removed the Isle Royale wolf as an example of when inbreeding depression is caused by few highly deleterious alleles.

Throughout: The word 'significant' is used throughout in reference to statistical results. I suggest changing this to 'statistically significant' to avoid confusion regarding biological versus statistical significance.

Changed accordingly throughout the manuscript, thanks for pointing this out.

191-193: This is not clearly written. Any reader paying attention will wonder what is the

'proportion of LOF mutations'? Is this the proportion of all LOF alleles where an individual has at least one copy? Or are you still referring to proportion of homozygous LOF alleles as above? Please define each metric explicitly in the results. As currently written, this lack of clarity makes it quite hard to not lose track of what has been done from one section to the next. Perhaps start out each section with a clear description of the relevant metric of load and why it was used.

This metrics was taking into account all LoF alleles, so both in homozygote and heterozygote state, but we can see that this was not very clear. We have now tried to clarify the metrics of load and why they were used. **Line 135-143, 183-187**

196: I think this should be figure S2a

Fixed, but now to fig S3a (**line 192**)

Section beginning at 204: It would help to point out here the caveat that we have no idea what are the fitness impact of any of these LOF alleles. There will always be alleles that satisfy the criteria used here to identify candidate LOF alleles, even when there are no fitness effects at all.

Good point, we have now pointed this out (**line 219-222**).

316: it is unclear what 'genomic composition' means.

Changed to: "The risk and impact of introducing deleterious alleles through gene flow has recently been debated within conservation genetics" (**line 328-329**)

338-342: This it true for segregating deleterious alleles. However, populations that are small for a long time are expected to have more fixed deleterious alleles (and often lower intrinsic fitness in one to one comparisons) than historically larger ones.

This is a very valid point, although recent studies have revealed that purging has occurred it does not take away the high need of gene flow for many small populations. We have tried to nuance our statements in this section. **Line 347-354**

Marty Kardos

Reviewer #4 (Remarks to the Author):

The authors addressed many aspects of the comments of reviewer 2. However, in my opinion there are still few adjustments to be made. In the attached file the comments are

added in blue to the former comments of reviewer 2.

Comment to comment 1:

I appreciate that 7 individuals could be added to the analyses of mutational load and that the rodent cycle could be considered in the linear regression. However, I would suggest to only include the regression model with the rodent cycle considered in the main text and in Figure 1. The outcome of the regression should be shown in detail in the supplementary file as it is done for the ancestry analyses. Litter size is not anymore significantly influenced by proportion of homozygous LoF genotypes when considering rodent cycle. This should be accounted for in the discussion (line 301-303).

We have now removed the regression model that was only a function of the proportion of mutations. The regression model that we are using now controls for the phase of the rodent cycle but also for sex, in accordance with the request from reviewer 1. We have replaced figure 1 and we have included the regression model outputs in the supplementary material.

The knowledge that ancestry influences the fitness parameters as shown in the ancestry analysis (with more individuals) suggests that it should be at least mentioned why you do not include the immigration ancestry in the analyses of mutational load or show the model with ancestry included in the supplementary file. This might imply to first show the analyses of ancestry and then the analyses of mutational load.

Thanks for this comment. We have not included individual ancestry as a parameter in our statistical analyses, since we now control for both the phase of the rodent cycle as well as individual sex (to make it more similar to the regression model used on the larger dataset, in accordance with reviewer 1). We do not find it meaningful to add a fourth explanatory variable to a dataset of ~20 datapoints. We have now mentioned this in the manuscript (**line 533-537**). If the reviewer would like, we could include ancestry in the analyses but in that case we believe that we should exclude either rodent phase or sex as explanatory variable to maintain some statistical power.

The confounding factor of inbreeding was is not yet addressed/discussed. See comments 5B below.

Comment to comment 5B:

I concur with reviewer 1 that a comparison of the effect size of the regression with the proportion of homozygous LoF genotypes with a regression of overall homozygosity from all markers or FROH would be appreciated at least in the supplementary file to get more insight into the difference in the effect of overall inbreeding, homozygous LoFs and homozygous missense genotypes. Please mention the correlation of FROH with the proportion of LoFs, missense alleles and synonymous alleles in homozygous state that

you mentioned in the answer to reviewer 1, also at least in the supplementary file in case you are already at the limit of number of words in the main text.

We have now included a correlation between genome wide heterozygosity per kb and synonymous, missense and LoF genotypes in homozygous state (**line 174-178 and 282-285, fig S2**) and find that it correlates strongly with synonymous and missense genotypes but also with LoF genotypes although more weakly.

Minor comments:

Line 143: It should be LoF genotypes in homozygous state

Fixed (**line 151**)

Line 174: correct spelling of mutations

This sentence is no longer present

Line 196: It should be fig. S2a

Fixed, but is now S3a (**line 192**)

Figure 2c: Shouldn't the legend of y-axis be "litter size" instead of "average litter size"?

Correct, changed accordingly.

REVIEWERS' COMMENTS

Reviewer #1 (Remarks to the Author):

I only have a few remaining very minor comments below. I think this is a really nice paper that is likely to be widely read.

Figure 1: add p-values to H and I

Tables S4-S17: The captions say that "...where bold values indicate significant results. Asterisks show level of significance as $P < 0.001$ ***, $P < 0.01$ ** and $P < 0.05$ *", but there are not statistically significant results in several of these tables. The captions would make more sense if you remove this part of the caption from the tables that do not show any statistically significant results.

296: Add 'putatively' before deleterious. We do not know what the fitness effects of individual mutations are.

316: It is unclear why "especially during years of high food abundance".

Reviewer #4 (Remarks to the Author):

My concerns have been fully addressed. I am looking forward to see the paper published adding important information to the knowledge of inbreeding depression.

REVIEWERS' COMMENTS

Reviewer #1 (Remarks to the Author):

I only have a few remaining very minor comments below. I think this is a really nice paper that is likely to be widely read.

Figure 1: add p-values to H and I

Done

Tables S4-S17: The captions say that "...where bold values indicate significant results. Asterisks show level of significance as $P < 0.001$ ***, $P < 0.01$ ** and $P < 0.05$ *", but there are not statistically significant results in several of these tables. The captions would make more sense if you remove this part of the caption from the tables that do not show any statistically significant results.

Good point, changed accordingly.

296: Add 'putatively' before deleterious. We do not know what the fitness effects of individual mutations are.

True, the word "putatively" has been added.

316: It is unclear why "especially during years of high food abundance".

We have changed this to that the effects of lowered lifetime reproductive success may be further enhanced by the fluctuating resources.

Reviewer #4 (Remarks to the Author):

My concerns have been fully addressed. I am looking forward to see the paper published adding important information to the knowledge of inbreeding depression.

Thank you!